# A Watermark for Order-Agnostic Language Models

**Ruibo Chen**,* **Yihan Wu**\*, **Yanshuo Chen, Chenxi Liu, Junfeng Guo, Heng Huang**[†]
Department of Computer Science
University of Maryland, College Park, MD, USA
`{rbchen,ywu42,cys,cxliu539,gjf2023,heng}@umd.edu`

## Abstract

Statistical watermarking techniques are well-established for sequentially decoded language models (LMs). However, these techniques cannot be directly applied to order-agnostic LMs, as the tokens in order-agnostic LMs are not generated sequentially. In this work, we introduce PATTERN-MARK, a pattern-based watermarking framework specifically designed for order-agnostic LMs. We develop a Markov-chain-based watermark generator that produces watermark key sequences with high-frequency key patterns. Correspondingly, we propose a statistical pattern-based detection algorithm that recovers the key sequence during detection and conducts statistical tests based on the count of high-frequency patterns. Our extensive evaluations on order-agnostic LMs, such as ProteinMPNN and CMLM, demonstrate PATTERN-MARK's enhanced detection efficiency, generation quality, and robustness, positioning it as a superior watermarking technique for order-agnostic LMs.

## 1 Introduction

With the rapid advancement in language models' capabilities and their increasing adoption, researchers and regulators have expressed growing concerns regarding their potential misuse in generating harmful content. Distinguishing between human-generated and AI-generated text has become a significant area of research. Statistical watermarking (Aaronson, 2022; Kirchenbauer et al., 2023a; Christ et al., 2023; Zhao et al., 2023; Liu et al., 2023a; Zhang et al., 2024; Huo et al., 2024; Chen et al., 2024b) offers a promising solution for identifying text produced by sequential LLMs (Brown, 2020; Touvron et al., 2023). This technique embeds a covert statistical signal into the generated content using a pseudo-random generator, which can later be detected through a statistical hypothesis test.

Order-agnostic LMs, where content is not generated in a left-to-right sequence (see Figure 1), have promising applications in fields such as protein generation (Dauparas et al., 2022; Alamdari et al., 2023), machine translation (Ghazvininejad et al., 2019; Kasai et al., 2020; Huang et al., 2022), speech generation (Chen et al., 2020; Peng et al., 2020), and time series forecasting (Chen et al., 2021; Tang & Matteson, 2021). Although watermarking techniques have proven successful in sequentially decoded LMs, most of these methods cannot be directly applied to order-agnostic LMs. This limitation arises because watermark generation and detection typically rely on previously generated context (n-gram), which is not consistently available in order-agnostic LMs.

In our work, we propose PATTERN-MARK, a pattern-based watermarking framework for order-agnostic language models (LMs). Our approach consists of a Markov-chain-based watermark generator and a rigorous statistical, pattern-based detection algorithm. The intuition of using a Markov chain is in order to enable recovery during detection, the pseudo-randomness of the watermark must be derived from the context, and the Markov chain is well-suited for establishing the dependencies within the context. Specifically, in the watermark generator, we utilize a Markov chain to generate a key sequence of length $n$, which is equal to the output content length. Each generated key is

---

*Equal Contributions
[†]This work was partially supported by NSF IIS 2347592, 2348169, DBI 2405416, CCF 2348306, CNS 2347617.

Figure 1: Sequentially decoding vs. Order-agnostic decoding. Sequentially decoded text, follows a fixed, left-to-right construction while order-agnostic decoding generates text by filling in words without adherence to traditional reading order. Most of current watermarking methods typically rely on previously generated context (n-gram), which is not consistently available in order-agnostic LMs.

then used as a random seed to modify the output distribution at the corresponding time step. This Markov-chain-based key sequence generation ensures a higher occurrence of certain key patterns compared to non-watermarked models. During detection, the key sequence can be recovered from the given content, allowing the use of a hypothesis test on the occurrence of specific key patterns to guarantee a controlled theoretical false positive rate.

Our contributions can be summarized as:

- We propose PATTERN-MARK, a pattern-based watermarking framework specifically designed for order-agnostic LMs. Our approach introduces a novel Markov-chain-based key sequence generation method and a rigorous statistical pattern-based watermark detection technique. To the best of our knowledge, this is the first work to explore watermarking for order-agnostic LMs, which are popular architectures in diverse real-world applications including machine translation, protein synthesis, speech generation, and time series forecasting.

- Through comprehensive experiments on two popular order-agnostic LMs, Protein-MPNN (Dauparas et al., 2022) and CMLM (Ghazvininejad et al., 2019), we demonstrate the superiority of PATTERN-MARK in terms of detection efficiency, generation quality, and robustness compared to baseline methods.

## 2 RELATED WORK

**Statistical watermarks.** Aaronson (2022) proposed the first statistical watermarking algorithm via Gumbel sampling. Kirchenbauer et al. (2023a) extended the statistical watermarking framework by splitting the model's tokens into red and green lists and promoting the use of green tokens. Huo et al. (2024) further improved the detectability of the red-green list watermark by employing learnable networks to optimize the separation of the red-green lists and the increment of green list logits. Zhao et al. (2023) introduced the unigram watermark, which enhances the robustness of statistical watermarking by leveraging one-gram hashing to generate watermark keys. Liu et al. (2023b) also contributed to the robustness of statistical watermarking by incorporating the semantics of the generated content as watermark keys. Liu et al. (2023a) proposed an unforgeable watermarking scheme that utilizes neural networks to modify token distributions. Christ et al. (2023); Kuditipudi et al. (2023); Hu et al. (2023a); Wu et al. (2023b) focused on distortion-free watermarking, which aims to preserve the original language model distribution. We will demonstrate that achieving distortion-free watermarking in order-agnostic LMs is not feasible using their approaches.

**Order-agnostic language models.** Order-agnostic LMs have gained significant attention in machine translation and other NLP tasks due to their ability to improve inference efficiency by generating multiple tokens simultaneously rather than sequentially. The pioneering work by Gu et al. (2017) introduced a non-sequential decoding approach through knowledge distillation, reducing machine translation latency and accelerating generation without sacrificing quality. Following this, Mask-Predict by Ghazvininejad et al. (2019) further advanced order-agnostic LMs by introducing a masked language model that iteratively refines predictions. Kasai et al. (2020) explored parallel decoding with a disentangled context transformer to achieve faster translation, focusing on improving both speed and quality. Other works, such as Chen et al. (2020); Peng et al. (2020) adapted the non-autoregressive transformer to accelerate speech generation tasks, and Chen et al. (2021); Tang & Matteson (2021) used spatial-temporal transformer and probabilistic-based transformer to improve the performance of time-series forecasting. Moreover, in the recent seminal work, Dauparas et al. (2022) proposed an order-agnostic language model for protein design. During generation, the decoding order is randomly

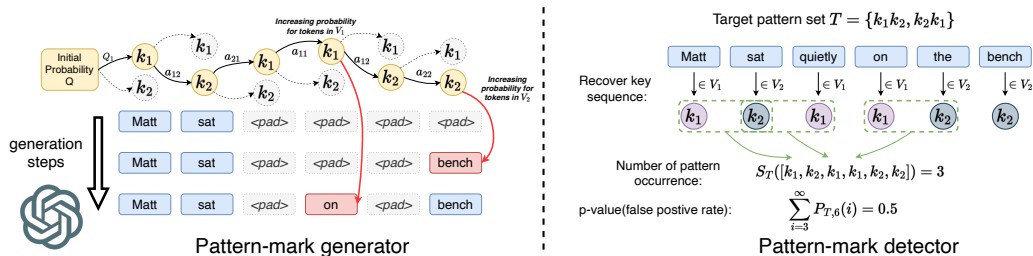

Figure 2: Illustration of PATTERN-MARK. The watermark generation process begins with a Markov chain-based key generator that produces a key sequence. This sequence is then used to modify token probabilities during language model sampling. In the watermark detection phase, the key sequence is recovered from the generated content, and the false positive rate is calculated by counting the occurrences of specific patterns within the key sequence.

sampled from the set of all possible permutations, then the protein sequence is generated from the given order. Order-agnostic decoding enables design in cases where, for example, the middle of the protein sequence is fixed and the rest needs to be designed, as in protein binder design where the target sequence is known.

## 3 PATTERN-MARK: A PATTERN-BASED WATERMARKING FRAMEWORK

### 3.1 PRELIMINARY

**Notations.** Let $V := \{t_1, \ldots, t_N\}$ denote the vocabulary (or token) set of a language model, with $N = |V|$ representing its size. Define $\mathcal{V}$ as the set of all sequences, including those of length zero. We use $P_M$ to denote the distribution of a language model and $P_W$ to denote the distribution of the watermarked language model. A sequentially decoded language model generates a token sequence conditioned on a given prompt. At any step in this process, the probability of generating the next token $x_n \in V$, conditioned on the preceding tokens from $x_1$ to $x_{n-1}$, is denoted by $P_M(x_n \mid x_1, x_2, \ldots, x_{n-1})$. In order-agnostic LMs, the model does not have access to the full preceding sequence $\boldsymbol{x}_{1:n-1}$, but it may have access to tokens from other positions. Therefore, we use $\boldsymbol{x}_n^{oa}$ to represent all tokens that the order-agnostic LM can observe when generating the $n$-th token. Under this setting, the probability of generating the next token $x_n \in V$ for order-agnostic LMs is given by $P_M(x_n \mid \boldsymbol{x}_n^{oa})$. In our watermarking framework, we will have a key set $K$, and the set of all key sequences is denoted by $K^*$. We denote $\boldsymbol{k}[1:n]$ as a key sequence of length $n$.

**Watermarking problem.** A language model service provider aims to watermark the generated content such that all other users can verify if the content is generated by the LM without needing access to the LM or the original prompt. A watermark framework primarily consists of two components: a *watermark generator* and a *watermark detector*. The watermark generator embeds a watermark into the text through a pseudo-random sampling strategy seeded by watermark keys. The watermark detector, on the other hand, detects the presence of the watermark within the content using a statistical hypothesis test.

**Discussion – Why current watermarking schemes can not be adapted to order-agnostic LMs.** Most of the current watermarking schemes (Kirchenbauer et al., 2023a; Liu et al., 2023b; Huo et al., 2024) rely on previously generated content as part of the watermark key for producing the next token. During detection, the previously generated content is extracted and used to verify the watermark. This approach works well in sequentially decoded LMs, where both generation and detection follow a left-to-right sequence. However, in order-agnostic LMs, content is generated in a non-sequential manner, and the detector does not know the order in which the tokens were generated, making it impossible to use previously generated content for watermark detection. For example, if we apply the Soft-watermark scheme (Kirchenbauer et al., 2023a) to order-agnostic LMs, when generating the $n$-th token, the watermark key is derived from the hash of the previous $i$ tokens. However, in order-agnostic LMs, some of these previous $i$ tokens may not yet be generated, resulting in missing or empty tokens. During detection, all previous $i$ tokens are expected to be present, and the absence of those tokens during generation causes a mismatch, making it impossible to retrieve the correct watermark key and thus failing to detect the watermark. To the best of our knowledge, the only watermarking scheme that can be applied to order-agnostic LMs is the Unigram (Zhao et al., 2023) (which is very similar to SelfHash Kirchenbauer et al. (2023b) without left tokens). However, Zhao

---

**Algorithm 1** PATTERN-MARK generator

---

1: **Input:** secret key sk, prompt $\boldsymbol{x}_{-m:0}$, generate length $n \in \mathbb{N}$, key space $K$, key space dimension $l$, transition matrix $A \in [0,1]^{l \times l}$, initial distribution $Q$, generation order $o$
2: First generate key sequence $\boldsymbol{k} \in K^n$: sample $\boldsymbol{k}[1]$ from the initial key distribution $Q$
3: **for** $i = 2, \ldots, n$ **do**
4:      Sample $\boldsymbol{k}[i]$ from $\Pr(\cdot | \boldsymbol{k}[i-1], A)$
5: **for** $i = 1, \ldots, n$ **do**
6:      Calculate the LM distribution for generating the $o_i$-th token $P_M(\cdot \mid \boldsymbol{x}_{-m:0}, \boldsymbol{x}_{o_i}^{oa})$.
7:      Calculate watermarked distribution $P_W(\cdot | \boldsymbol{x}_{-m:0}, \boldsymbol{x}_{o_i}^{oa}, \boldsymbol{k}[i])$ via Eq. 1.
8:      Sample the next token $\boldsymbol{x}_{o_i}$ using distribution $P_W(\cdot \mid \boldsymbol{x}_{-m:0}, \boldsymbol{x}_{o_i}^{oa})$.
9: **return** $\boldsymbol{x}_{1:n}$.

---

**Algorithm 2** PATTERN-MARK detector

---

1: **Input:** key space $K = \{k_1, \ldots, k_l\}$, input sequence $\boldsymbol{x} = \{x_1, \ldots, x_n\}$, vocabulary split $\{V_1, \ldots, V_l\}$, FPR threshold $f$, pattern length $m$, target patterns $T \subset K^m$, pattern occurrence probability $P_{T,n}$
2: Initialize the pattern occurrence number: $c = 0$.
3: Initialize an empty key sequence: $\boldsymbol{k}$.
4: **for** $i = 1, \ldots, n$ **do**
5:      Recover the i-th key based on the current token $x_i$: if $x_i \in V_c$, $\boldsymbol{k}[i] = k_c$,
6: **for** $i = m, \ldots, n$ **do**
7:      **if** $\boldsymbol{k}[i-m+1:i] \in T$ **then**
8:          $c = c + 1$
9: Calculating $P_{T,n}$ from Alg. 3
10: Calculating the false positive probability for the input text $\text{FPR}_{\boldsymbol{x}} = \sum_{i=c}^{n-m+1} P_{T,n}(i)$
11: **if** $\text{FPR}_{\boldsymbol{x}} \leq f$ **then**
12:      **Return:** $\boldsymbol{x}$ is watermarked by PATTERN-MARK
13: **else**
14:      **Return:** $\boldsymbol{x}$ is **not** watermarked by PATTERN-MARK

---

et al. (2023) uses a fixed red-green list to split the vocabulary and consistently promote the probability of green tokens, which will hurt the generation quality across the entire generation process.

**Discussion – Why distortion-free watermarking schemes can not be adapted to order-agnostic LMs.** We claim that the current distortion-free scheme cannot be applied to order-agnostic LMs. There are two primary reasons for this: 1) The distortion-free property relies on the autoregressive nature of LMs. According to Christ et al. (2023); Hu et al. (2023a); Wu et al. (2024), a distortion-free watermark requires independent probabilities $P_W(x_n | \boldsymbol{x}_n^{oa}, k_n)$ for every token $n$. However, some order-agnostic LMs are non-autoregressive. For example, $x_{n-1}$ and $x_n$ are generated simultaneously from a joint distribution $P_W(x_n, x_{n-1} | \boldsymbol{x}_n^{oa}, k_n, k_{n-1})$. This makes it impossible to maintain independent $P_W(x_n | \boldsymbol{x}_n^{oa}, k_n)$ for each $n$. 2) The distortion-free property also requires non-repeating watermark keys during generation. In sequentially decoded LMs, the watermark key space can be expanded using n-gram hashing. However, this approach is not feasible in order-agnostic LMs because some tokens in the n-gram may not yet be generated, making it impossible to fully reconstruct the n-gram during watermark detection.

### 3.2 PATTERN-BASED WATERMARKING FRAMEWORK

In this work, we propose a pattern-based watermarking framework, specifically designed for order-agnostic LMs (see Figure 2 for a detailed illustration). Given $l$ keys $K := \{k_1, k_2, \ldots, k_l\}$, we partition the vocabulary $V$ into $l$ distinct parts $V_1, V_2, \ldots, V_l$, where each part corresponds to a unique key pattern. During the generation process, we first produce a sequence of keys $\boldsymbol{k}_{i_{1:n}} := k_{i_1}, k_{i_2}, \ldots, k_{i_n}$, where $n$ is the length of the generated content. When generating the $j$-th token, we will promote the probability of tokens within $V_{i_j}$. We follow the probability promotion strategy in Kirchenbauer et al. (2023a). Specifically, given an initial LM token probability $P_M(t)$ and a key

---

**Algorithm 3** Compute pattern occurrence probability under the null hypothesis

1: **Input:** key space $K$, input sequence length $n$, pattern length $m$, $m > 1$, target patterns $T \in K^m$, key space dimension $l$
2: **Output:** pattern occurrence probability $P_{T,n}$
3: Initialize the status probability distribution $P_n(C, \boldsymbol{S})$, where $C \in \{0, \ldots, n-m+1\}$ represents the number of occurrences of patterns, $\boldsymbol{S} \in K^{m-1}$ represents the last $m-1$ keys in a key sequence.
4: Initialize $P_i(C = c, \boldsymbol{S} = \boldsymbol{s}) = 0, \forall i, c, \boldsymbol{s}$
5: $P_{m-1}(C = 0, \boldsymbol{S} = \boldsymbol{s}) = (\frac{1}{l})^{m-1}, \forall \boldsymbol{s}$
6: **for** $i = m, \ldots, n$ **do**
7:     **for** $\forall \boldsymbol{s} \in K^{m-1}$ **do**
8:         $\boldsymbol{s}' = [k, \boldsymbol{s}_{1:m-2}]$
9:         **for** $c = 0, \ldots, i - m$ **do**
10:             $P_i(C = c, \boldsymbol{S} = \boldsymbol{s}) += \frac{1}{l} \sum_{k \in K, [k, \boldsymbol{s}] \notin T} P_{i-1}(C = c, \boldsymbol{S} = \boldsymbol{s}')$

11:         **for** $c = 1, \ldots, i - m + 1$ **do**
12:             $P_i(C = c, \boldsymbol{S} = \boldsymbol{s}) += \frac{1}{l} \sum_{k \in K, [k, \boldsymbol{s}] \in T} P_{i-1}(C = c - 1, \boldsymbol{S} = \boldsymbol{s}')$

13: $P_{T,n}(c) = \sum_{\boldsymbol{s} \in K^{m-1}} P_n(C = c, \boldsymbol{S} = \boldsymbol{s})$
14: **return** $P_{T,n}$.

---

$k_i$, the watermarked probability for the token, denoted by $P_W(t)$, is formulated as:

$$P_W(t|k_i) = \begin{cases} \dfrac{e^\delta P_M(t)}{\sum_{t' \notin V_i} P_M(t') + \sum_{t' \in V_i} e^\delta P_M(t')}, & t \in V_i \\ \dfrac{P_M(t)}{\sum_{t' \notin V_i} P_M(t') + \sum_{t' \in V_i} e^\delta P_M(t')}, & t \notin V_i; \end{cases} \quad (1)$$

During detection, we analyze the generated sequence to uncover regular patterns in the key sequence, verifying the presence of the watermark by identifying these patterns.

**Markov-chain-based key sequence generation.** During watermark detection, we do not have access to the entire key sequence used in watermarking. To address this, we propose a Markov-chain-based key sequence generation method, which generates a key sequence with high-frequency patterns (i.e., subsequences). This allows us to focus on detecting specific patterns.

A Markov process is a stochastic process that describes a sequence of events, where the probability of each event depends only on the state of the previous event. Given a transition matrix $A := (a_{ij})_{l \times l}$, where $a_{ij}$ denotes the transition probability from state $k_i$ to $k_j$, we define the Markov-chain-based key sequence generation as follows:

**Definition 3.1** (Markov-chain based key sequence generation). *Let $X_t$ be the random variable representing the $t$-th position of the sequence, $A = (a_{ij})_{l \times l}$ the transition matrix, and $Q = (q_1, ..., q_l)$ the initial probability. The key sequence is generated according to the rule $\Pr(X_1 = k_j) = q_j$ and $\Pr(X_{t+1} = k_j \mid X_t = k_i) = a_{ij}, t \geq 1$.*

For example, if $l = 2$ (i.e., we have two keys) and the transition matrix is $A = \begin{bmatrix} 0.3 & 0.7 \\ 0.7 & 0.3 \end{bmatrix}$, i.e., the next state has a probability of $0.3$ to remain the same and $0.7$ to switch to the other state. Then the number of patterns like $k_1 k_2$ and $k_2 k_1$ will be significantly higher than patterns like $k_1 k_1$ and $k_2 k_2$ in the generated Markov chain. This enables us to build a statistical detection algorithm based on the difference in the frequency of patterns between watermarked and non-watermarked content.

Besides, the self-transition probabilities $a_{11}$ and $a_{22} \in [0, 1]$ ($0.3$ in the example above) control the frequency of patterns in the generated key sequences. When $a_{11}$ and $a_{22}$ increase, patterns $k_1 k_1$ and $k_2 k_2$ are more likely to appear in the generated key sequence. Conversely, when $a_{11}$ and $a_{22}$ decrease, we are more likely to observe patterns $k_1 k_2$ and $k_2 k_1$. Therefore, to improve detection efficiency, we typically select either extremely large or small values for $a_{11}$ and $a_{22}$. However, with

large values of $a_{11}$ and $a_{22}$, the sequence may mainly consist of a single key, which could negatively impact generation quality. Hence, we generally select smaller values of $a_{11}$ and $a_{22}$ for key sequence generation.

**Statistical pattern-based detection.** During watermark detection, we recover a key sequence from the generated content. We use a hypothesis test to conduct watermark detection, where the null hypothesis $H_0$ states that the text is not watermarked with PATTERN-MARK. Under the null hypothesis, the recovered key sequence should follow a Markov chain with uniform transition probabilities (i.e., $A = (a_{ij} = 1/l)_{l \times l}$).

Given a set of patterns $T \subset K^m$, let $S_T(\boldsymbol{k}) := \sum_{j=1}^{n-m+1} \mathbf{1}_{\boldsymbol{k}[j:j+m-1] \in T}$, which represents the number of selected patterns in $T$ that appear in the key sequence $\boldsymbol{k} \in K^n$. We can compute its probability density function $P_{T,n}(i) := \Pr(S_T(\boldsymbol{k}) = i)$, where $\boldsymbol{k}$ is a random key sequence selected from $K^n$. Given an observed pattern count $c$, the p-value (false positive rate, a.k.a. FPR) for this hypothesis test is $\sum_{i=c}^{\infty} P_{T,n}(i)$. The detailed algorithms for watermark generation and detection are provided in Alg. 1 and Alg. 2. In Alg. 3, we provide the algorithm we used to calculate the probability density function $P_{T,n}$.

## 3.3 $P_{T,n}$ CALCULATION.

In this part, we provide a detailed analysis and explanation of the algorithm (Alg. 3) designed to compute the probability distribution of pattern occurrences in sequences generated by a Markov chain. The algorithm calculates the probability $P_{T,n}(c)$ that a set of target patterns $T \subseteq K^m$ occurs exactly $c$ times in a sequence of length $n$ over a key space $K$.

**Overview.** Our algorithm employs dynamic programming to efficiently compute the probability distribution by tracking the number of pattern occurrences and the last $m-1$ keys in the sequence. The key components including the key space $K$, sequence Length $n$, pattern length $m$, target patterns $T \subseteq K^m$.

**State Representation.** At each position $i$ in the sequence, the algorithm maintains a probability distribution over states. $C \in \{0, \ldots, n-m+1\}$: The number of occurrences of the target patterns so far. $\boldsymbol{S} \in K^{m-1}$: The last $m-1$ keys in the current sequence. The state probability distribution at position $i$ is denoted as $P_i(C = c, \boldsymbol{S} = \boldsymbol{s}) := \Pr(S_T(\boldsymbol{k}[1:i]) = c, \boldsymbol{k}[i-m+2:i] = \boldsymbol{s})$, where $\boldsymbol{k}$ is a random key sequence.

**Initialization.** The algorithm initializes the state probabilities with $P_i(C = c, \boldsymbol{S} = \boldsymbol{s}) = 0, \quad \forall i, c, \boldsymbol{s}$, and $P_{m-1}(C = 0, \boldsymbol{S} = \boldsymbol{s}) = (\frac{1}{l})^{m-1}, \forall \boldsymbol{s} = [s_1, \ldots, s_{m-1}] \in K^{m-1}$. Here, $P_{m-1}(C = 0, \boldsymbol{S} = \boldsymbol{s})$ represents the probability of starting with sequence $\boldsymbol{s}$ without any occurrences of the target patterns.

**Dynamic programming iteration.** For each position $i$ from $m$ to $n$ and each possible sequence $\boldsymbol{s} = [s_1, s_2, \ldots, s_{m-1}] \in K^{m-1}$, the algorithm updates the state probabilities through the following step: Construct the previous sequence $\boldsymbol{s}'$ by shifting $\boldsymbol{s}$ left and adding a new key $k \in K$: $\boldsymbol{s}' = [k, s_1, s_2, \ldots, s_{m-2}]$. **Case 1:** If $[\boldsymbol{s}', s_{m-1}] \notin T$ (no new occurrence of a target pattern), update for $c = 0$ to $i - m$: $P_i(C = c, \boldsymbol{S} = \boldsymbol{s}) += \frac{1}{l} P_{i-1}(C = c, \boldsymbol{S} = \boldsymbol{s}')$. **Case 2:** If $[\boldsymbol{s}', s_{m-1}] \in T$ (new occurrence of a target pattern), update for $c = 1$ to $i - m + 1$: $P_i(C = c, \boldsymbol{S} = \boldsymbol{s}) += \frac{1}{l} P_{i-1}(C = c - 1, \boldsymbol{S} = \boldsymbol{s}')$.

**Final Probability Computation.** After completing the iterations up to position $n$, the total probability for each count $c$ is calculated by summing over all possible sequences $\boldsymbol{S}$: $P_{T,n}(c) = \sum_{\boldsymbol{s} \in K^{m-1}} P_n(C = c, \boldsymbol{S} = \boldsymbol{s})$. The time complexity of this algorithm is $O(n^2 l^m)$ and the memory complexity is $O(n^2 l^{m-1})$.

## 4 EXPERIMENTS

Our experimental section consists of three parts. In the first part, we compare the detection efficiency of PATTERN-MARK with the baseline. In the second part, we evaluate the generation quality of PATTERN-MARK. In the third part, we assess the robustness of the PATTERN-MARK when subjected to random token modification and paraphrasing attacks. We focus on two seq2seq tasks in our experiments: machine translation with CMLM model (Ghazvininejad et al., 2019), and protein generation with ProteinMPNN (Dauparas et al., 2022).

Table 1: Empirical true positive rate (TPR) for watermark detection on protein generation, each row is averaged over around 800 generated protein sequences. We report the TPR under {10%, 1%, 0.1%, 0.01%, 0.001%} theoretical FPR. A more detailed comparison can be found in Table 5.

| Protein generation | pLDDT | TPR@FPR= | | | | |
|---|---|---|---|---|---|---|
| | | 10% | 1% | 0.1% | 0.01% | 0.001% |
| No Watermark | 85.46 | - | - | - | - | - |
| Soft watermark($\delta = 1.50$) | 84.30 | 12.99 | 2.68 | 0.80 | 0.13 | 0.00 |
| Multikey($\delta = 1.50$) | 84.20 | 67.20 | 33.87 | 14.32 | 6.96 | 2.68 |
| Unigram($\delta = 1.50$) | 83.58 | 96.12 | 92.10 | 85.14 | 77.51 | 69.75 |
| PATTERN-MARK($\delta = 0.50$) | 85.42 | 47.26 | 17.94 | 6.29 | 2.54 | 0.67 |
| PATTERN-MARK($\delta = 0.75$) | 85.05 | 81.79 | 51.00 | 30.66 | 17.80 | 9.37 |
| PATTERN-MARK($\delta = 1.00$) | 84.74 | 97.46 | 88.35 | 68.27 | 50.87 | 38.02 |
| PATTERN-MARK($\delta = 1.25$) | 84.50 | 99.87 | 98.80 | 96.12 | 87.28 | 77.24 |
| PATTERN-MARK($\delta = 1.50$) | 84.02 | **100.00** | **100.00** | **99.87** | **99.06** | **96.39** |

Table 2: Empirical true positive rate (TPR) for watermark detection on machine translation, each row is averaged over around 1000 generated sequences. We report the TPR under {10%, 1%, 0.1%, 0.01%, 0.001%} theoretical FPR. A more detailed comparison can be found in Table 6.

| Machine translation | BLEU | TPR@FPR= | | | | |
|---|---|---|---|---|---|---|
| | | 10% | 1% | 0.1% | 0.01% | 0.001% |
| No Watermark | 32.85 | - | - | - | - | - |
| Soft watermark($\delta = 5$) | 25.58 | 97.91 | 85.04 | 60.22 | 36.89 | 20.44 |
| Multikey($\delta = 5$) | 27.82 | 89.93 | 60.52 | 32.00 | 13.66 | 5.38 |
| Unigram($\delta = 5$) | 22.23 | 99.90 | 99.20 | 95.41 | 88.53 | 78.07 |
| PATTERN-MARK($\delta = 1$) | 33.02 | 19.74 | 4.59 | 0.50 | 0.10 | 0.00 |
| PATTERN-MARK($\delta = 2$) | 32.60 | 57.33 | 28.41 | 13.96 | 8.47 | 4.39 |
| PATTERN-MARK($\delta = 3$) | 31.67 | 86.04 | 65.70 | 46.86 | 33.30 | 24.73 |
| PATTERN-MARK($\delta = 4$) | 29.43 | 98.21 | 89.53 | 79.16 | 67.00 | 56.83 |
| PATTERN-MARK($\delta = 5$) | 24.23 | **100.00** | **99.80** | **98.01** | **95.41** | **91.13** |

## 4.1 EXPERIMENTAL SETTINGS

In this part, we provide a general introduction of our experimental settings, a more detailed version can be found at Appendix B.

Since there are generally no watermarking methods specifically designed for order-agnostic LMs, we mainly compare PATTERN-MARK with three baseline methods adapted from existing approaches: the Soft watermark Kirchenbauer et al. (2023a), Unigram Zhao et al. (2023), and Multikey watermark Kirchenbauer et al. (2023a). We mainly use three metrics to measure the quality of the generated content. Specifically, we use pLDDT (Jumper et al., 2021) and protein diversity to measure the quality of the protein generation task, and BLEU (Papineni et al., 2002) to measure the quality of the machine translation task.

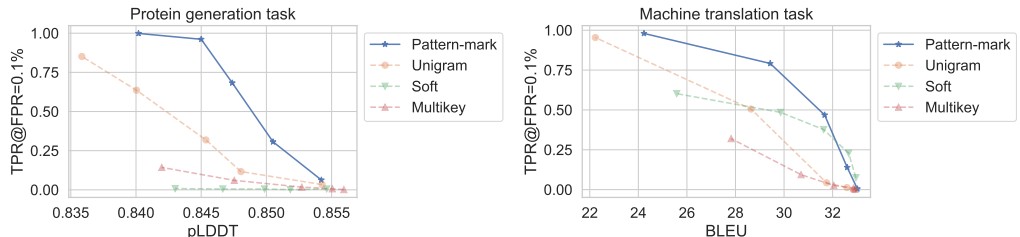

Figure 3: Comparison of the trade-off between TPR and generalization quality on protein generation and machine translation tasks. **Left.** TPR@FPR=0.1% vs. pLDDT on protein generation task. **Right.** TPR@FPR=0.1% vs. BLEU on machine translation task.

Table 3: Empirical true positive rate of different watermarks under random token modification with attack strength $\epsilon$ on protein generation task. Each row is averaged over around 800 generated protein sequences. We report TPR under 0.1% FPR.

| Protein generation | $\epsilon = 0$ | $\epsilon = 0.05$ | $\epsilon = 0.1$ | $\epsilon = 0.2$ | $\epsilon = 0.3$ |
|---|---|---|---|---|---|
| Soft watermark($\delta = 1.50$) | 0.80 | 0.54 | 0.27 | 0.13 | 0.00 |
| Multikey($\delta = 1.50$) | 14.32 | 10.84 | 9.24 | 5.22 | 2.01 |
| Unigram($\delta = 1.50$) | 85.14 | 82.73 | 78.05 | 68.41 | 54.08 |
| PATTERN-MARK($\delta = 0.50$) | 6.29 | 5.09 | 4.02 | 2.14 | 1.07 |
| PATTERN-MARK($\delta = 0.75$) | 30.66 | 24.63 | 19.01 | 11.24 | 5.22 |
| PATTERN-MARK($\delta = 1.00$) | 68.27 | 60.78 | 51.41 | 31.46 | 16.73 |
| PATTERN-MARK($\delta = 1.25$) | 96.12 | 90.09 | 83.13 | 64.79 | 39.63 |
| PATTERN-MARK($\delta = 1.50$) | **99.87** | **98.66** | **97.05** | **87.82** | **65.73** |

Table 4: Empirical true positive rate of different watermarks under ChatGPT paraphrasing attack with attack strength $\epsilon$ on machine translation task. Each row is averaged over around 1000 generated sequences. We report TPR under 0.1% FPR.

| Machine translation | $\epsilon = 0$ | $\epsilon = 0.05$ | $\epsilon = 0.1$ | $\epsilon = 0.2$ | $\epsilon = 0.3$ |
|---|---|---|---|---|---|
| Soft watermark($\delta = 5$) | 60.22 | 48.26 | 38.38 | 28.22 | 19.04 |
| Multikey($\delta = 5$) | 32.00 | 19.74 | 13.76 | 8.67 | 4.99 |
| unigram($\delta = 5$) | 95.41 | 90.03 | 87.14 | 77.27 | 64.01 |
| PATTERN-MARK($\delta = 1$) | 0.50 | 0.80 | 0.50 | 0.30 | 0.40 |
| PATTERN-MARK($\delta = 2$) | 13.96 | 13.26 | 11.57 | 9.97 | 7.38 |
| PATTERN-MARK($\delta = 3$) | 46.86 | 42.47 | 38.88 | 34.20 | 29.41 |
| PATTERN-MARK($\delta = 4$) | 79.16 | 74.68 | 72.58 | 64.81 | 58.13 |
| PATTERN-MARK($\delta = 5$) | **98.01** | **97.10** | **95.81** | **93.32** | **89.53** |

**Watermark settings.** In PATTERN-MARK, we select the key set $K = \{k_1, k_2\}$, the Markov-chain transition matrix $A = [[0, 1], [1, 0]]$, and the initial distribution $Q = [0.5, 0.5]$. The key patterns are defined as $T = \{k_1 k_2 k_1 \ldots, k_2 k_1 k_2 \ldots\}$, where $k_1$ and $k_2$ appear alternately, $T \subset K^m$. Under this configuration, the probability $P_{T,n}$ can be calculated with Algorithm 4, which optimizes the process described in Algorithm 3. The computational complexity is reduced from $O(n^2 2^m)$ to $O(n^2 m)$.

## 4.2 DETECTION EFFICIENCY

We evaluate the detectability of our watermark on machine translation and protein generation. For protein generation tasks, the generated content length is around 300-500, and for machine translation tasks, the generated content length is around 50-100. We select $\delta \in \{0.5, 0.75, 1.0, 1.25, 1.5\}$ for protein generation task, and $\delta \in \{1, 2, 3, 4, 5\}$ for machine translation task. We report the true positive rate at the theoretical false positive rate (TPR@FPR) with 10%, 1%, 0.1%, 0.01%, and 0.001% FPR. In Appendix Table 5 and Table 6 we present the more detailed results.

The results are presented in Table 1 and Table 2. As shown, under the same probability promotion level $\delta$, our method consistently outperforms the baseline methods. Additionally, we include the average pLDDT and BLEU scores in Table 1 and Table 2. When $\delta = 1.25$, PATTERN-MARK surpasses all baselines in both generation quality and detectability in the protein generation task. Moreover, when $\delta = 4$, PATTERN-MARK outperforms the Soft watermark in both generation quality and detectability in the machine translation task, and at $\delta = 5$, PATTERN-MARK surpasses the Unigram method on both metrics for the same task.

## 4.3 QUALITY-DETECTABILITY TRADE-OFF

We examine the trade-off between watermark detection efficiency and generation quality of our method across two tasks: for the protein generation task, we utilize pLDDT as a quality metric, while for the machine translation task, we employ the BLEU score. We also made additional quality comparisons in Appendix C.1.

Figure 3 illustrates the trade-off between watermark detectability and generalization quality across both tasks. Each point in the figure corresponds to an experiment with a specific $\delta$, where the values

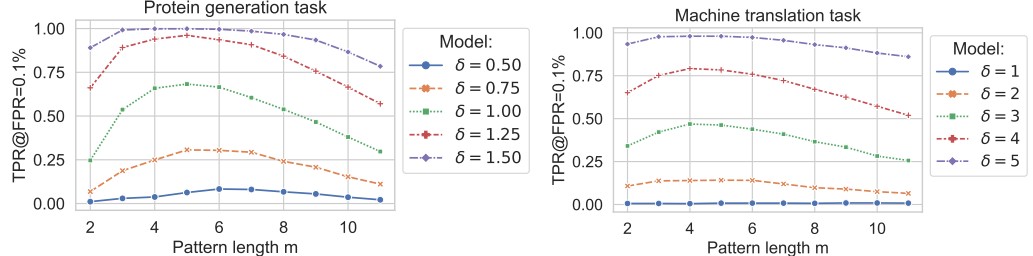

Figure 4: Evaluation of the effect of pattern length $m$ on the detection efficiency of PATTERN-MARK. **Left.** Protein generation task. **Right.** Machine translation task.

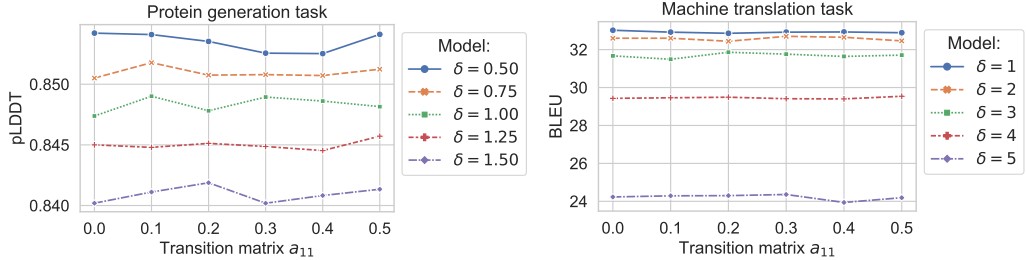

Figure 5: Evaluation of the effect of transition matrix probability on the generation quality of PATTERN-MARK. **Left.** Protein generation task. **Right.** Machine translation task.

of $\delta$ are determined according to Sec. 4.2. The results indicate that our method achieves a generally superior trade-off compared to all baselines.

## 4.4 ROBUSTNESS

We compare the robustness of PATTERN-MARK against the baselines on both the protein generation and machine translation tasks. For the protein generation task, we employ a random token modification attack, while for the machine translation task, we use a paraphrase attack generated via ChatGPT. Unfortunately, we are unable to evaluate paraphrase attacks on protein generation due to the absence of a suitable rephrasing model. To perform the robustness experiment, we reuse the generated content from Sec. 4.2. The results of these experiments are presented in Table 3 and Table 4. From these tables, it is evident that PATTERN-MARK consistently outperforms the baselines across both tasks.

## 4.5 ABLATION STUDY

We present an ablation study to check the effect of pattern length $m$ and the transition matrix on the detectability and the quality of our watermark.

**Pattern Length $m$.** We analyzed the detection efficiency of PATTERN-MARK across various pattern lengths $m$ in protein and language contexts. As shown in Figure 4, the efficiency initially improves with an increase in $m$ then subsequently declines. Specifically, for protein generation tasks, the optimal detection performance is attained at $m = 5$. In contrast, for machine translation tasks, the best performance is achieved with $m = 4$.

Generally, increasing the pattern length $m$ leads to improved detection efficiency to some extent. This improvement can be attributed to that longer patterns are less likely to appear in a randomly generated key sequence. However, excessively long patterns also introduce a higher sensitivity to errors during key sequence recovery. Consequently, while longer patterns enhance detection, they also risk increasing the false positive rate due to their sensitivity to minor errors. This trade-off results in the observed decline in detection efficiency for excessively long patterns.

**Transition matrix.** We evaluated the quality of content generated using various configurations of the transition matrix $A = \begin{bmatrix} a_{11} & 1 - a_{11} \\ 1 - a_{11} & a_{11} \end{bmatrix}$. According to Figure 5, the quality of the generated content remains consistent across different values of $a_{11}$. Consequently, we select $a_{11} = 0$, which corresponds to the strongest watermarking signal, as it does not compromise the quality while enhancing the watermark's detectability.

## 5 CONCLUSION

In conclusion, PATTERN-MARK presents a significant advancement in language model watermarking, specifically addressing the challenges posed by watermarking order-agnostic LMs. By innovatively leveraging a Markov-chain-based key sequence generation and a statistical pattern-based detection algorithm, this framework effectively embeds and detects statistical watermarks in non-sequentially generated text. Through comprehensive evaluation of prominent non-sequential models like ProteinMPNN and CMLM, PATTERN-MARK has proven to be superior in terms of detection accuracy and reliability compared to traditional watermarking techniques, providing a novel approach for watermarking the order-agnostic LMs.

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

# A    ALGORITHMS FOR CALCULATING PATTERN OCCURRENCE PROBABILITY

In Alg. 4, we show the exact algorithm we used for calculating the pattern occurrence probabilities.

---

**Algorithm 4** Compute pattern occurrence probability under the null hypothesis

---

1: **Input:** key space $K = \{k_1, k_2\}$, input sequence length $n$, pattern length $m$, $m > 2$, target patterns $T_m = \{\underbrace{k_1 k_2 \cdots}_{\text{length } m}, \underbrace{k_2 k_1 \cdots}_{\text{length } m}\} \in K^m$ ($k_1, k_2$ appear alternately), initial key distribution $Q = \{0.5, 0.5\}$.

2: **Output:** pattern occurrence probability $P_{T,n}$

3: Initialize the status probability distribution $P_n(C, M)$, where $C \in \{0, \dots, n - m + 1\}$ represents the number of occurrences of patterns, given the current key sequence $\boldsymbol{k}$, $M := \min(\arg\max_{m'}\{\boldsymbol{k}[-m' : -1] \in T_{m'}\}, m - 1)$ represents the longest tail sequence (at most $m - 1$) where $k_1$ and $k_2$ appear alternately.

4: Initialize $P_i(C = c, M = m') = 0, \forall i, c, m'$

5: $P_1(C = 0, M = 1) = 1$

6: **for** $i = 2, \dots, m - 1$ **do**

7:     $P_i(C = 0, M = 1) = \frac{1}{2} \sum\limits_{m'=1}^{i} P_{i-1}(C = 0, M = m')$

8:     **for** $m' = 2, \dots, i$ **do**

9:         $P_i(C = 0, M = m') = \frac{1}{2} P_{i-1}(C = 0, M = m' - 1)$

10: **for** $i = m, \dots, n$ **do**

11:     **for** $c = 0, \dots, i - m$ **do**

12:         $P_i(C = c, M = 1) = \frac{1}{2} \sum\limits_{m'=1}^{m-1} P_{i-1}(C = c, M = m')$

13:         **for** $m' = 2, \dots, m - 1$ **do**

14:             $P_i(C = c, M = m') += \frac{1}{2} P_{i-1}(C = c, M = m' - 1)$

15:     **for** $c = 1, \dots, i - m + 1$ **do**

16:         $P_i(C = c, M = m - 1) += \frac{1}{2} P_{i-1}(C = c - 1, M = m - 1)$

17: $P_{T,n}(c) = \sum\limits_{m'=1}^{m-1} P_n(C = c, M = m')$

18: **return** $P_{T,n}$.

---

# B    EXPERIMENTAL SETTINGS

**Baseline.** Since there are generally no watermarking methods specifically designed for order-agnostic LMs, we mainly compare PATTERN-MARK with three baseline methods adapted from existing approaches: the Soft watermark Kirchenbauer et al. (2023a), Unigram Zhao et al. (2023), and Multikey watermark Kirchenbauer et al. (2023a). For Unigram, we adhere to the configurations as originally specified in the literature. In the case of the Soft watermark, we utilize 1-gram tokens as watermark keys. When this 1-gram token is an empty token, we just use this empty token directly as the seed for watermark embedding. The Multikey watermark method modifies the Soft watermark approach by substituting the n-gram watermark keys with two predetermined keys, randomly selecting one to seed the watermarking process at each generation step.

**Metrics.** We mainly use three metrics to measure the quality of the generated content. Specifically, we use pLDDT and protein diversity to measure the quality of the protein generation task, and BLEU to measure the quality of the machine translation task.

- **pLDDT.** The pLDDT (Jumper et al., 2021) metric is a crucial evaluation tool in the field of computational biology, particularly in protein structure prediction. This metric quantitatively assesses the confidence in the predicted local structure of proteins at the residue level. Developed as part of the AlphaFold2 architecture by DeepMind, pLDDT has become a standard for evaluating the reliability of predicted protein structures.

- **Protein diversity.** Protein diversity is an entropy-based metric to quantify the diversity of amino acid sequences within a set of proteins, where a higher entropy value indicates a greater variety of amino acids at each position in the sequence, indicating higher protein diversity.

- **BLEU score.** For the machine translation task, we utilize the BLEU score (Papineni et al., 2002) to evaluate the lexical similarity between translations produced by the machine and those generated by humans.

**Settings.** In our experiments, we adhere to the default parameters set for ProteinMPNN (Dauparas et al., 2022) and CMLM (Ghazvininejad et al., 2019). ProteinMPNN is specifically developed for protein redesign tasks, requiring the generation of a protein sequence from a given 3D protein backbone structure. It employs an encoder-decoder architecture, enabling order-agnostic sequence generation. This is facilitated by the encoded structural context which guides the subsequent decoding steps. We utilize the v_48_020 model checkpoint for ProteinMPNN.

The CMLM operates under an encoder-encoder framework. During inference, the model processes in fixed steps, simultaneously generating multiple tokens at each step irrespective of their sequential positions. Tokens with lower probabilities are identified for regeneration in subsequent steps. For the CMLM, we apply it to a Romanian-English translation task using the available pre-trained checkpoint.

**Dataset.** For CMLM, we use the data collected from news crawl[1] *news.2015.ro.shuffled* whose length is larger than 128 to encourage longer generation. The filtered dataset has 1003 samples.

For ProteinMPNN, we use the protein features from PCSB Protein Data Bank[2], which is published from 2020 Jan. 1st to 2023 Dec. 31st. We limit the number of polymer residues per deposited model to between 400 and 500. The filtered dataset has 747 samples.

In PATTERN-MARK, we select the key set $K = \{k_1, k_2\}$, the Markov-chain transition matrix $A = \begin{bmatrix} 0 & 1 \\ 1 & 0 \end{bmatrix}$, and the initial distribution $Q = [0.5, 0.5]$. The key patterns are defined as $T = \{k_1 k_2 k_1 \ldots, k_2 k_1 k_2 \ldots\}$, where $k_1$ and $k_2$ appear alternately, $T \subset K^m$. Under this configuration, the probability $P_{T,n}$ can be calculated using Algorithm 4, which optimizes the process described in Algorithm 3. The computational complexity of Algorithm 4 is $O(n^2 m)$, compared to $O(n^2 2^m)$ for Algorithm 3.

In Section 4.5, we present an ablation study to evaluate the effects of different transition matrices $A$ and various lengths of key patterns $m$ on detection accuracy.

# C ADDITIONAL EXPERIMENTS

In this part, we show additional experiments about the detection efficiency comparison between PATTERN-MARK and baselines on protein generation and machine translation tasks. We select $\delta \in \{0.5, 0.75, 1.0, 1.25, 1.5\}$ for protein generation task, and $\delta \in \{1, 2, 3, 4, 5\}$ for machine translation task.

The results are presented in Table 5 and Table 6. As shown, under the same probability promotion level $\delta$, our method consistently outperforms the baseline methods.

## C.1 QUALITY EVALUATION

We compare the performance of our method across two tasks: for the protein generation task, we utilize pLDDT as a quality metric and protein diversity as a measure of diversity, while for the machine translation task, we employ the BLEU score. Additionally, we examine the trade-off between watermark detection efficiency and generation quality.

---

[1]https://data.statmt.org/news-crawl/
[2]https://www.rcsb.org/

Table 5: Empirical true positive rate (TPR) for watermark detection on protein generation, each row is averaged over around 1000 generated protein sequences. We report the TPR under {10%, 1%, 0.1%, 0.01%, 0.001%} theoretical FPR.

| | pLDDT | TPR@FPR= | | | | |
|---|---|---|---|---|---|---|
| | | 10% | 1% | 0.1% | 0.01% | 0.001% |
| No Watermark | 85.46 | - | - | - | - | - |
| Soft watermark($\delta = 0.50$) | 85.46 | 6.83 | 1.07 | 0.54 | 0.13 | 0.00 |
| Soft watermark($\delta = 0.75$) | 85.18 | 9.24 | 1.74 | 0.13 | 0.13 | 0.00 |
| Soft watermark($\delta = 1.00$) | 84.99 | 11.24 | 1.87 | 0.54 | 0.00 | 0.00 |
| Soft watermark($\delta = 1.25$) | 84.67 | 12.58 | 2.28 | 0.54 | 0.13 | 0.00 |
| Soft watermark($\delta = 1.50$) | 84.30 | 12.99 | 2.68 | 0.80 | 0.13 | 0.00 |
| Multikey($\delta = 0.50$) | 85.59 | 1.20 | 0.53 | 0.13 | 0.13 | 0.13 |
| Multikey($\delta = 0.75$) | 85.51 | 4.95 | 1.07 | 0.67 | 0.40 | 0.13 |
| Multikey($\delta = 1.00$) | 85.27 | 16.47 | 4.69 | 1.74 | 0.80 | 0.54 |
| Multikey($\delta = 1.25$) | 84.75 | 44.18 | 13.79 | 6.02 | 2.28 | 1.07 |
| Multikey($\delta = 1.50$) | 84.20 | 67.20 | 33.87 | 14.32 | 6.96 | 2.68 |
| Unigram($\delta = 0.50$) | 85.43 | 10.58 | 5.09 | 3.35 | 2.54 | 1.34 |
| Unigram($\delta = 0.75$) | 84.80 | 33.33 | 18.21 | 11.65 | 8.17 | 5.76 |
| Unigram($\delta = 1.00$) | 84.54 | 68.54 | 44.44 | 31.86 | 21.55 | 17.27 |
| Unigram($\delta = 1.25$) | 84.00 | 88.09 | 78.05 | 63.59 | 47.79 | 37.22 |
| Unigram($\delta = 1.50$) | 83.58 | 96.12 | 92.10 | 85.14 | 77.51 | 69.75 |
| PATTERN-MARK($\delta = 0.50$) | 85.42 | 47.26 | 17.94 | 6.29 | 2.54 | 0.67 |
| PATTERN-MARK($\delta = 0.75$) | 85.05 | 81.79 | 51.00 | 30.66 | 17.80 | 9.37 |
| PATTERN-MARK($\delta = 1.00$) | 84.74 | 97.46 | 88.35 | 68.27 | 50.87 | 38.02 |
| PATTERN-MARK($\delta = 1.25$) | 84.50 | 99.87 | 98.80 | 96.12 | 87.28 | 77.24 |
| PATTERN-MARK($\delta = 1.50$) | 84.02 | 100.00 | 100.00 | 99.87 | 99.06 | 96.39 |

Table 6: Empirical true positive rate (TPR) for watermark detection on machine translation, each row is averaged over around 1000 generated sequences. We report the TPR under {10%, 1%, 0.1%, 0.01%, 0.001%} theoretical FPR.

| | BLEU | TPR@FPR= | | | | |
|---|---|---|---|---|---|---|
| | | 10% | 1% | 0.1% | 0.01% | 0.001% |
| No Watermark | 32.85 | - | - | - | - | - |
| Soft watermark($\delta = 1$) | 32.96 | 54.94 | 23.13 | 7.48 | 2.59 | 0.70 |
| Soft watermark($\delta = 2$) | 32.66 | 79.46 | 45.36 | 22.83 | 10.67 | 4.39 |
| Soft watermark($\delta = 3$) | 31.63 | 87.74 | 61.91 | 37.69 | 18.25 | 7.38 |
| Soft watermark($\delta = 4$) | 29.86 | 93.82 | 75.17 | 48.45 | 27.12 | 12.66 |
| Soft watermark($\delta = 5$) | 25.58 | 97.91 | 85.04 | 60.22 | 36.89 | 20.44 |
| Multikey($\delta = 1$) | 32.92 | 5.08 | 0.80 | 0.20 | 0.10 | 0.00 |
| Multikey($\delta = 2$) | 32.86 | 14.46 | 2.49 | 1.00 | 0.50 | 0.50 |
| Multikey($\delta = 3$) | 32.05 | 34.30 | 8.87 | 2.59 | 1.00 | 0.60 |
| Multikey($\delta = 4$) | 30.70 | 60.62 | 25.42 | 9.37 | 3.49 | 1.40 |
| Multikey($\delta = 5$) | 27.82 | 89.93 | 60.52 | 32.00 | 13.66 | 5.38 |
| Unigram($\delta = 1$) | 32.89 | 2.79 | 0.50 | 0.00 | 0.00 | 0.00 |
| Unigram($\delta = 2$) | 32.59 | 17.65 | 4.89 | 1.30 | 0.50 | 0.20 |
| Unigram($\delta = 3$) | 31.75 | 56.73 | 26.02 | 10.47 | 4.29 | 1.99 |
| Unigram($\delta = 4$) | 28.64 | 91.03 | 72.28 | 50.55 | 31.01 | 18.94 |
| Unigram($\delta = 5$) | 22.23 | 99.90 | 99.20 | 95.41 | 88.53 | 78.07 |
| PATTERN-MARK($\delta = 1$) | 33.02 | 19.74 | 4.59 | 0.50 | 0.10 | 0.00 |
| PATTERN-MARK($\delta = 2$) | 32.60 | 57.33 | 28.41 | 13.96 | 8.47 | 4.39 |
| PATTERN-MARK($\delta = 3$) | 31.67 | 86.04 | 65.70 | 46.86 | 33.30 | 24.73 |
| PATTERN-MARK($\delta = 4$) | 29.43 | 98.21 | 89.53 | 79.16 | 67.00 | 56.83 |
| PATTERN-MARK($\delta = 5$) | 24.23 | 100.00 | 99.80 | 98.01 | 95.41 | 91.13 |

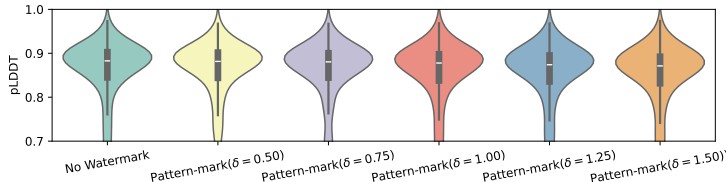

Figure 6: Comparison of the pLDDT of the protein generated from PATTERN-MARK with the non-watermarked model.

Table 7: Comparison of the protein diversity of the protein generated from different watermarking approaches.

|  | 1-gram entropy | 2-gram entropy | 3-gram entropy |
|---|---|---|---|
| No Watermark | 4.091 | 8.019 | 11.830 |
| Soft watermark($\delta = 1.50$) | 4.024 | 7.828 | 11.554 |
| Multikey($\delta = 1.50$) | 4.034 | 7.912 | 11.700 |
| Unigram($\delta = 1.50$) | 3.934 | 7.708 | 11.371 |
| PATTERN-MARK($\delta = 1.50$) | 4.105 | 7.998 | 11.763 |

In Figure 6 and Table 7, we compare the quality of protein generation between PATTERN-MARK and the non-watermarked model. It is evident that under the same $\delta$, our method consistently produces proteins of high quality and diversity.

## D BROADER IMPACT STATEMENT

Machine learning models have profound impacts across various domains, demonstrating significant potential in both enhancing efficiencies and addressing complex challenges (Yang et al., 2020; 2019; Wen et al., 2023; Chakraborty et al., 2022; Cai et al., 2022; Chen et al., 2024a; Xu & Li, 2017; Liu et al., 2024; Wen et al., 2024) However, alongside these positive impacts, there are concerns about the integrity and security of machine learning applications (Hong et al., 2024; Hu et al., 2023b; Wang et al., 2023b;a; Wu et al., 2022; 2023a;c). Watermarking emerges as a pivotal technique in this context. It ensures the authenticity and ownership of digital media, and also can help people to distinguish AI-generated content.

This paper presents a new watermarking technique, PATTERN-MARK, specifically designed for order-agnostic LMs. This advancement addresses the critical challenge of embedding and detecting watermarks in LMs that do not generate content sequentially, such as those used in protein design and machine translation. The introduction of PATTERN-MARK could significantly enhance the security and integrity of generated content across various fields, fostering trust in AI-generated content and enabling robust detection of AI authorship. Such a watermarking approach is pivotal for regulatory compliance, intellectual property protection, and preventing the misuse of generative models, especially in sensitive applications like medical research and international communication. Additionally, PATTERN-MARK's ability to maintain high generation quality while ensuring robust watermark detection contributes to the practical deployment of secure, order-agnostic LMs in commercial and research settings, promoting broader acceptance and integration of these advanced AI technologies.

