# OpenReview forum: "A Watermark for Order-Agnostic Language Models"
_ICLR.cc/2025/Conference — ICLR 2025 Poster_

### Official Review · Reviewer_4ffB · 2024-10-29

**Soundness:** 2
**Presentation:** 3
**Contribution:** 2
**Rating:** 5
**Confidence:** 3

**Summary:**

This paper proposes a method for generating and detecting watermarks in ordered-agnostic language models (LMs). At a high level, the method is an extension of the red-green list approach in which the vocabulary is divided into two sets: the "green" list and the "red" list. The detector counts the number of tokens in the green list as a statistic for detection. The current method takes a more sophisticated approach to generating patterns in the key sequence by utilizing a Markov model during the text generation process. Detection is based on the number of patterns that appear in the key sequence, which are reconstructed from the text and the vocabulary partitions. The authors demonstrate that this method is more effective in terms of detectability and text quality than other existing watermarking schemes.

**Strengths:**

The paper presents a thorough investigation of watermarking for order-agnostic language models, a topic that appears to be less explored in the existing literature. The proposed method introduces several new ideas, including using Markov models to generate more complex patterns, setting it apart from previous approaches. The new approach shows promising performance when compared to existing methods. Additionally, the authors have explored the impact of various parameters on the method's effectiveness.

**Weaknesses:**

The assertion that "this is the first work to explore watermarking for order-agnostic language models" is somewhat exaggerated. The same problem can be addressed using existing methods, even though they may not be specifically tailored for order-agnostic settings (e.g., Zhao et al., 2023).

Additionally, the proposed method is more computationally intensive compared to the red-green list approach. Furthermore, the Markov structure seems to introduce another factor (in addition to $\delta$), which could cause the distribution of the watermarked language model to differ from that of the original model.

**Questions:**

1. My main question is why the Markov model is suitable for order-agnostic language models (LMs). Order-agnostic LMs do not generate tokens from left to right, while Markov models create sequences in a left-to-right manner. This presents a discrepancy between the order-agnostic nature of the LMs and the order-dependent nature of Markov models. It is unclear why Markov models would be particularly appropriate for order-agnostic LMs. Additionally, the discussions in Section 3.2, central to the proposed procedure, are vague and require more careful explanations.

2. The discussion regarding why distortion-free watermarking schemes cannot be adapted to order-agnostic LMs is misleading. For instance, on page 3, line 196, it states, “A distortion-free watermark requires independent probabilities…” The term “independent probabilities” is unclear in this context. Distortion-free refers to $P_W(x_n|\mathbf{x}_n^{oa},k_n) = P_M(x_n|\mathbf{x}_n^{oa},k_n)$. Additionally, the assertion that “The distortion-free property also requires non-repeating watermark keys during generation” lacks clarity. The authors should be cautious with such an impossibility claim, as it requires a proper justification.

3. The Markov structure seems to introduce another factor (in addition to $\delta$), which could cause the distribution of the watermarked language model to differ from that of the original model. I wonder if there is any way to quantify the distortion caused by the Markov model.

4. A potential drawback of the current approach is its time complexity. Could the authors report the computational time compared to other methods?

---

> ### Author Response · Authors · 2024-11-13
> **Response to Reviewer 4ffB (1/2)**
>
> Thank you for the valuable feedback on our manuscript. We truly appreciate the time and effort that you have invested in evaluating our work. Below, we address each of your concerns:
>
> > W1: The assertion that "this is the first work to explore watermarking for order-agnostic language models" is somewhat exaggerated. The same problem can be addressed using existing methods, even though they may not be specifically tailored for order-agnostic settings (e.g., Zhao et al., 2023).
>
> A: We are deeply sorry for the confusion. In this statement, we want to show that we are the first to “study the order-agnostic LM watermarking problem” instead of “designing the first watermarking approach for order-agnostic LM”. We are also aware that some of the existing watermarking methods (including Zhao et al., 2023) can be adapted to order-agnostic models and we have included them as baselines, however, the experimental evidence shows that those existing approaches do not perform well on order-agnostic LMs.
>
> > W2: Additionally, the proposed method is more computationally intensive compared to the red-green list approach.
>
> A: We show the exact time cost for the protein generation task below.
>
> |                |  Generation  |   Detection   |
> |----------------|:------------:|:-------------:|
> | No Watermark   | 0.69s/sample |      -       |
> | Soft Watermark | 0.79s/sample | 0.058s/sample |
> | Multikey       | 0.74s/sample | 0.046s/sample |
> | Unigram        | 0.73s/sample | 0.024s/sample |
> | Pattern-mark   | 0.74s/sample | 0.021s/sample |
>
> For the generation phase, experiments were conducted on a single RTX 6000 Ada GPU. The primary computational overhead is due to model inference, while the time required to apply watermarks is minimal. Our proposed method does not incur additional computation compared to other baseline methods.
>
> For the detection phase, experiments were run on a CPU. We only need to pre-calculate the false positive rates table defined in Algorithm 4 once and store the results for use in Algorithm 3. The pre-calculation is also efficient, taking approximately only 7 seconds.
>
> As the results indicate, our method does not introduce additional computational complexity in practical applications.
>
> > W3: Furthermore, the Markov structure seems to introduce another factor (in addition to $\delta$), which could cause the distribution of the watermarked language model to differ from that of the original model.
>
> Could you please clarify which specific factor raised your concern?
>
> In our experiments, distribution bias is measured by the degradation of BLEU or pLDDT scores. It’s important to note that distribution bias and watermark strength involve a trade-off: stronger watermarking reduces output quality, leading to greater distribution divergence from the original model (i.e., more distribution difference from the original model or higher distribution bias).
>
> As shown in Figure 3, at the same watermark strength, our method achieves higher BLEU or pLDDT scores, indicating that our approach introduces less distribution bias compared to baseline methods.
>
> > Q1 a) why the Markov model is suitable for order-agnostic language models? Order-agnostic LMs do not generate tokens from left to right, while Markov models create sequences in a left-to-right manner. This presents a discrepancy between the order-agnostic nature of the LMs and the order-dependent nature of Markov models.
>
> In sequentially decoded LMs, the watermark key depends on preceding tokens, which are typically unavailable in order-agnostic models. To address this, we use a Markov model to generate watermark keys. The Markov chain, therefore, functions solely as a watermark key generator, which is independent of the order-agnostic nature of the language models. We also discuss the intuition of using the Markov chain in Lines 45-49 of our work.
>
> To elaborate further, during watermark detection, the only input available is the generated sequence, and one important challenge is that we cannot access the generation order at this stage. However, we still need to capture the pseudo-randomness of the watermark (e.g., the seed for the random number generator).
>
> Some methods, like Unigram and Multikey, rely on globally consistent information (e.g., fixed green list tokens). Our experiments show that these methods suffer from low output quality or insufficient watermark strength.
>
> Another approach commonly used is injecting this information into the context. For instance, in Soft watermarking, the seed is derived from the history n-gram (the last n tokens). However, for order-agnostic language models, the last n tokens may not have been generated yet, as the generation order is unknown. To address this issue, our method generates a key sequence using the Markov chain in a left-to-right manner and uses these keys to inject information into the corresponding tokens. This creates correlations between neighboring tokens, which can then be utilized to detect our watermark.

---

> ### Author Response · Authors · 2024-11-13
> **Response to Reviewer 4ffB (2/2)**
>
> > Q1 b)  the discussions in Section 3.2, central to the proposed procedure, are vague and require more careful explanations.
>
> We sincerely apologize for this and will revise it to include more details. If you have any questions about our procedure, we would be more than happy to provide further explanations and address any concerns.
>
>
> > Q2 a): The discussion regarding why distortion-free watermarking schemes cannot be adapted to order-agnostic LMs is misleading. For instance, on page 3, line 196, it states, “A distortion-free watermark requires independent probabilities…” The term “independent probabilities” is unclear in this context.
>
> A: We are deeply sorry for the confusion. There are generally two levels of distortion-free: 1) token level distortion-free, i.e.,  $P_W(x_n|x_{n}^{oa},k_n)=P_M(x_n|x_{n}^{oa})$ and 2) sentence-level distortion-free, i.e., $P_W(x_{1:n}|x_{1:n}^{oa},k_{1:n})=E_{k_{1:n}}[\prod_{i=1}^nP_M(x_{i}|x_{i}^{oa})]$. The distortion-freeness discussed in our paper is sentence-level distortion-free instead of token-level distortion-free. In order to achieve sentence-level distortion-free, the current distortion-free watermarking schemes require $P_W(x_{1:n}|x_{1:n}^{oa},k_{1:n})=\prod_{i=1}^nP_W(x_{i}|x_{i}^{oa},k_{i})$, which requires the independence of $P_W(x_{i}|x_{i}^{oa},k_{i})$, $i=1,...,n$. For example, In section 4.2.1 of Hu et al. (2023a)[2], the authors claim “It is crucial that $E_i$ values are independent to ensure the unbiased nature of the entire sequence, rather than just the single-token generation process.” Here, the independence of $E_i$ is equal to the “independent probabilities $P_W(x_n|x_{n}^{oa}, k_n)$ for each step $n$”.
>
> > Q2 b): Additionally, the assertion that “The distortion-free property also requires non-repeating watermark keys during generation” lacks clarity. The authors should be cautious with such an impossibility claim, as it requires a proper justification.
>
> A: As described in the response to Q2 a), the independence of $P_W(x_{i}|x_{i}^{oa},k_{i})$, $i=1,...,n$ are required for sentence-level distortion-freeness. In order to achieve such independence, we need the watermark keys $k_{i}$ to be non-repeating. For example, in section 4.2.2 of Hu et al. (2023a)[2], the authors claim “$E_i$ are independent with each other if only their context codes are different”. Here, the context code refers to the watermark keys. We will add the detailed discussion in our paper to support the claims.
>
>
> > Q3: The Markov structure seems to introduce another factor (in addition to
> ), which could cause the distribution of the watermarked language model to differ from that of the original model. I wonder if there is any way to quantify the distortion caused by the Markov model.
>
> A:   Please refer to the response for W3, as the concerns are the same.
>
> > Q4: A potential drawback of the current approach is its time complexity. Could the authors report the computational time compared to other methods?
>
> A:  Please refer to the response for W2, as the concerns are the same.
>
> Thank you again for your thoughtful feedback, and we would be happy to discuss further if you have any additional questions or concerns.
>
>
> [1] Zhao, Xuandong, et al. "Provable robust watermarking for ai-generated text." ICLR 2024.
> [2] Hu, Zhengmian, et al. "Unbiased watermark for large language models." ICLR 2024.

---

> > ### Comment · Reviewer_4ffB · 2024-11-18
> >
> > I appreciate the authors for their detailed responses to my questions. However, a key question remains unanswered: why is it necessary to use a Markov chain for order-agnostic language models (LMs)? As acknowledged by the authors, it is possible to utilize existing methods—without resorting to a Markov chain or other dependent processes—to generate the watermark keys. The usage of the Markov chain does not seem essential for addressing the order-agnostic nature of LMs. I am unclear why the authors promote their approach as a solution for order-agnostic LMs, considering it primarily serves as a generalization of the green-red list approach to accommodate more general patterns.
> >
> > Regarding W3, my question is whether the Markov structure, particularly the dependence among the watermark keys, can cause the distribution of the watermarked LM to differ from that of the original model. Since the dependence structure of the Markov model is chosen independent of the underlying order-agnostic LMs, would a particular choice of this dependence structure lead to more distortion than other choices?
> >
> > In their response to Question 2, the authors state that "distortion-freeness" refers to sentence-level distortion-freeness. Does this imply that existing methods can still be used if we focus solely on token-level distortion freeness? The authors mention that the current distortion-free watermarking schemes require $P_W(x_{1:n} \vert \mathbf{x}_{1:n}^{oa}, k _ {1:n} )= \prod_i P_W(x_i \vert \mathbf{x}_i^{oa},k_i)$. Why? Shouldn't $P_W$ on the RHS be replaced by $P_M$ (otherwise, how can you link the watermarked LM with the original LM)? If this is the case, the LHS and RHS are for different LMs, and the interpretation the authors provided would not hold.
> >
> > Additionally, the term "independent probabilities" is somewhat confusing; we typically refer to random variables as being independent rather than probabilities (probabilities are non-random). Overall, I find the discussions quite confusing.
> >
> > I would also appreciate it if the authors could discuss the potential pitfalls of their approaches. I believe there may not be a free lunch, and counting more intricate patterns might be less effective in certain scenarios.

---

> ### Author Response · Authors · 2024-11-18
> **Response to the Follow-Up Questions (1/3)**
>
> Thank you for your timely feedback and valuable comments. We are glad to have addressed some of your concerns and would like to provide clarification on the remaining questions below:
>
>
> > Follow-up Q1: I appreciate the authors for their detailed responses to my questions. However, a key question remains unanswered: why is it necessary to use a Markov chain for order-agnostic language models (LMs)? As acknowledged by the authors, it is possible to utilize existing methods—without resorting to a Markov chain or other dependent processes—to generate the watermark keys. The usage of the Markov chain does not seem essential for addressing the order-agnostic nature of LMs. I am unclear why the authors promote their approach as a solution for order-agnostic LMs, considering it primarily serves as a generalization of the green-red list approach to accommodate more general patterns.
>
> **TL,DR:** Current watermarking methods are not suitable for order-agnostic LM watermarking because of the watermark key mismatch during the watermark generation and detection processes. To address this problem, we use a markov chain to generate the watermark keys to avoid such mismatch.
>
> First, we want to clarify that we do not claim that the Markov chain is necessary for order-agnostic LM watermarking. Instead, it is one of the promising solutions for order-agnostic LM watermarking. To further explain our motivation for employing the Markov chain, we begin by discussing the challenges associated with applying existing methods to order-agnostic LMs. We then demonstrate the advantages of our proposed method.
>
> Challenges for existing watermarking methods:
>
> - We acknowledge that some existing methods, such as Unigram and Multikey, can be applied to order-agnostic language models; these methods are also included as our baselines. However, their reliance on globally consistent information, without incorporating contextual information, poses a significant limitation that can severely impact performance. For instance, Unigram employs a fixed green list for all tokens, leading to substantial distribution distortion compared to the unwatermarked model, as tokens across all positions are biased toward the same fixed green list. As shown in Tables 5 and 6, this results in degraded output quality. Multikey, on the other hand, can be viewed as an extension of Unigram. It randomly selects one green list from a predefined pool, introducing variability while still maintaining global consistency. Although Multikey reduces distortion and improves output quality compared to Unigram, its watermark strength diminishes significantly, making detection more challenging.
>
> - A more common approach in sequential models [1,2,3,4] is to use n-gram history (preceding n tokens) as watermark keys to generate the green list for the next token, as employed by the baseline method Soft Watermark. For instance, as illustrated in Figure 1, when predicting the last token "*bench*" in the sentence "*Matt sat quietly on the bench*",  the method utilizes the preceding 2-gram "*on the*" as the watermark key to construct the green list. This approach generates more diverse green lists, leading to generally better performance compared to Unigram and Multikey in sequential models. During detection, the watermark detector can use the same preceding tokens to recover the green list and identify the existence of watermarks. However, in order-agnostic LMs, some of these preceding n tokens may not yet be generated, resulting in missing or empty tokens. For example, when generating the last token "*bench*", the preceding 2-gram may not yet be available.  During detection, all previous i tokens are expected to be present, and the absence of those tokens during generation causes a mismatch, making it impossible to retrieve the correct watermark key and thus failing to detect the watermark.
>
> Advantages for using Markov chain in order-agnostic LM watermarking:
>
> - By utilizing a Markov chain, we generate a watermark key for each position, with each key being correlated to the previous ones. In this case, watermark keys rely on the preceding watermark keys instead of the preceding generated content. This method guarantees that our green lists are more diverse than those produced by Unigram, while also eliminating the issue of watermark key mismatch that arises in the n-gram scheme.

---

> > ### Author Response · Authors · 2024-11-18
> > **Response to the Follow-Up Questions (2/3)**
> >
> > > Follow-up Q2: Regarding W3, my question is whether the Markov structure, particularly the dependence among the watermark keys, can cause the distribution of the watermarked LM to differ from that of the original model. Since the dependence structure of the Markov model is chosen independent of the underlying order-agnostic LMs, would a particular choice of this dependence structure lead to more distortion than other choices?
> >
> > Our method does not exhibit greater distortion under the same watermark strength, i.e., under the same true positive rate and false positive rate, our method introduces less distortion to the distribution.
> >
> > We acknowledge that under the same $\delta$ in Equation (1), our method causes more distribution distortion than Soft Watermark and Multikey. This is because our approach imposes stricter constraints on key patterns and green lists, and it is expected. This observation is also reported in Figure 3, Table 5, and Table 6. However, we also demonstrate that under the same $\delta$, baseline methods exhibit weaker watermark strength compared to ours, i.e.,  for the same $\delta$ and false positive rate, baseline methods produce lower true positive rates. We argue that this is due to a trade-off between watermark strength and distortion: a stronger watermark typically introduces greater distortion.
> >
> > A practical and fair comparison should be based on the same watermark strength, i.e., the same true positive rate at a given false positive rate. Under such conditions, our method demonstrates better output quality, as verified in Figure 3, indicating that it induces less distortion. In conclusion, our method achieves superior output quality and less distortion when evaluated at the same watermark strength.
> >
> >
> > > Follow-up Q3: In their response to Question 2, the authors state that "distortion-freeness" refers to sentence-level distortion-freeness. Does this imply that existing methods can still be used if we focus solely on token-level distortion freeness?
> >
> > We believe the existing methods still cannot be used to order agnostic LMs. Regardless of the level of distortion-freeness, the existing distortion-free watermarking techniques face the same challenge as other watermarking algorithms when applied to order-agnostic language models (LMs): the watermark key is derived from the hash of the preceding tokens [1,2]. This approach can result in empty tokens in order-agnostic LMs, causing mismatches in watermark keys during detection.
> >
> > Furthermore, as noted in [1], token-level distortion-freeness still leads to performance degradation because the overall sentence distribution deviates from that of the original LM. Since LM performance on downstream tasks is typically evaluated using sentence-level metrics (e.g., PPL, BLEU, and BERTScore) rather than token-level metrics, this mismatch can negatively impact the LM's effectiveness.
> >
> > > Follow-up Q4: The authors mention that the current distortion-free watermarking schemes require PW(x1:n|x1:noa,k1:n)=∏iPW(xi|xioa,ki). Why? Shouldn't PW on the RHS be replaced by PM (otherwise, how can you link the watermarked LM with the original LM)? If this is the case, the LHS and RHS are for different LMs, and the interpretation the authors provided would not hold.
> >
> > We are sorry for the confusion. A token-level distortion-free watermark is sentence-level distortion free requiring $P_W(x_{1:n}|x_{1:n}^{oa},k_{1:n}) = \prod_{i=1}^nP_W(x_{i}|x_{i}^{oa},k_{i})$. Recall that token-level distortion-free satisfy $E_{k_{i}}[P_W(x_{i}|x_{i}^{oa},k_{i})] = P_M(x_{i}|x_{i}^{oa})$, we can easily get $E_{k_{1:n}} [P_W(x_{1:n}|x_{1:n}^{oa},k_{1:n})] = P_M(x_{1:n}|x_{1:n}^{oa})$ when $P_W(x_{1:n}|x_{1:n}^{oa},k_{1:n}) = \prod_{i=1}^nP_W(x_{i}|x_{i}^{oa},k_{i})$. As current distortion-free watermarking schemes are sentence-level distortion free, they should satisfy $P_W(x_{1:n}|x_{1:n}^{oa},k_{1:n}) = \prod_{i=1}^nP_W(x_{i}|x_{i}^{oa},k_{i})$, which require the independence of each generation step ,i.e., $X_{i}\sim P_W(x_{i}|x_{i}^{oa},k_{i})$ and $X_{j}\sim P_W(x_{j}|x_{j}^{oa},k_{j})$ should be independent of each other when $i \neq j$.
> >
> > > Follow-up Q5: Additionally, the term "independent probabilities" is somewhat confusing; we typically refer to random variables as being independent rather than probabilities (probabilities are non-random). Overall, I find the discussions quite confusing.
> >
> > We agree, let $X_{i}\sim P_W(x_{i}|x_{i}^{oa},k_{i})$ and $X_{j}\sim P_W(x_{j}|x_{j}^{oa},k_{j})$, we want to claim sentence-level distortion-freeness can be satisfied when $X_{i}$ and $X_{j}$ are mutually independent for arbitrary $i\neq j$.

---

> > > ### Author Response · Authors · 2024-11-18
> > > **Response to the Follow-Up Questions (3/3)**
> > >
> > > > Follow-up Q6: I would also appreciate it if the authors could discuss the potential pitfalls of their approaches. I believe there may not be a free lunch, and counting more intricate patterns might be less effective in certain scenarios.
> > >
> > > One potential drawback of our method is that the pattern length $m$ can reduce robustness against attacks such as paraphrasing or random token replacement. This is because our pattern-based approach relies on $m$ tokens during detection, compared to only one token for Unigram and Multikey, and two tokens for Soft Watermark (the generated token itself and one preceding token as context).
> > >
> > > However, experimental results in Tables 3 and 4 demonstrate that our method consistently produces a stronger watermark, suggesting that the loss in robustness is not significant. Additionally, the pattern length $m$ can be adjusted to shorter spans to enhance robustness against such attacks, providing flexibility based on specific application needs.
> > >
> > >
> > > Thank you again. We are more than happy to address any additional questions or concerns you may have.
> > >
> > > [1] Hu, Zhengmian, et al. "Unbiased watermark for large language models." ICLR 2024.
> > > [2] Christ et al. “Undetectable watermark for large language models.” COLT 2024.
> > > [3] Kirchenbauer, John, et al. "A watermark for large language models." ICML 2023.
> > > [4] Kirchenbauer, John, et al. "On the reliability of watermarks for large language models." ICLR 2024.

---

> > > > ### Comment · Reviewer_4ffB · 2024-11-18
> > > >
> > > > Thank you for the clarification. I have some final comments.
> > > >
> > > > In the n-gram approach, the watermark is generated using previous tokens, which enables the natural use of contextual and semantic information during the watermark generation process. In contrast, the current method relies only on previously generated watermark keys without considering the content being produced. Therefore, contextual information is missing from the current procedure.
> > > >
> > > > A natural extension of the n-gram approach in the current context seems to directly use the previously generated n tokens $x^{oa} _ {1:n}$ (rather than the first $n$ tokens $x_{1:n}$) to determine the $n+1$th watermark key. It would be interesting to compare it with the proposed method.
> > > >
> > > > The authors never discussed the stationarity of the Markov model. If the model is non-stationary, the probability distribution could change over time, potentially leading to instability.

---

> ### Author Response · Authors · 2024-11-19
> **Response to the Follow-Up Questions (Round 2)**
>
> Thank you for your feedback and we truly appreciate the effort and attention you put into reviewing our comments. Below we answer your additional questions:
>
>
> >Follow-up Q7: In the n-gram approach, the watermark is generated using previous tokens, which enables the natural use of contextual and semantic information during the watermark generation process. In contrast, the current method relies only on previously generated watermark keys without considering the content being produced. Therefore, contextual information is missing from the current procedure.
>
>
> The n-gram approach does not utilize contextual or semantic information. Instead, n-gram tokens are hashed into seeds for a random number generator, which then generates the corresponding green list (e.g., Algorithm 2 in Kirchenbauer et al. (2023) [1] and Algorithm 1 in Kirchenbauer et al. (2024) [2]). This process relies solely on token IDs, without considering their semantic meaning, as confirmed by their official implementation [3]. Furthermore, preceding n-gram tokens have no positive influence in this approach because the green list is randomly generated by the random number generator. Therefore, our method does not have this limitation.
>
>
> >Follow-up Q8: A natural extension of the n-gram approach in the current context seems to directly use the previously generated n tokens $x^{oa} _ {1:n}$ (rather than the first $n$ tokens $x_{1:n}$) to determine the $n+1$ th watermark key. It would be interesting to compare it with the proposed method.
>
>
> Unfortunately, this extension does not work. The key problem is that during detection, the only input is the whole generated sentence, and the generation order is not known. As a result, it is impossible to identify the correct $x^{oa}_{i}$ and reconstruct the green list for detection.
>
>
> Referring to the example in Figure 1 (right figure), if $n=2$, the token "*bench*" ($x_i = \text{"bench"}$) is generated based on the two previously generated tokens "*Matt*" and "*sat*" ($x_i^{oa} = [\text{"Matt"}, \text{"sat"}]$). These tokens are used to generate the green list for "*bench*". However, during detection, the input is only the complete sentence "*Matt sat quietly on the bench*", and the generation order is unavailable. Consequently, we cannot determine which tokens were used to generate the green list for "*bench*". Without this information, detecting the watermark is not feasible, and therefore, this method fails.
>
>
> >Follow-up Q9: The authors never discussed the stationarity of the Markov model. If the model is non-stationary, the probability distribution could change over time, potentially leading to instability.
>
>
> Thank you for pointing this out. In fact, we have considered stationarity in our design. As detailed in our main experiments and analysis of the transition matrix (Sec. 4.5, Lines 470–476), the initial state distribution is $Q=[0.5, 0.5]$, and the transition matrix is defined as $A=[[a_{11},1-a_{11}],[1-a_{11},a_{11}]]$. When we compute $QA$, we obtain $QA=[0.5 a_{11}+0.5(1-a_{11}),0.5(1-a_{11})+0.5 a_{11}]=[0.5,0.5]=Q$, demonstrating that Q is a stationary distribution. We will incorporate the discussion of stationarity into our manuscript.
>
> [1] Kirchenbauer, John, et al. "A watermark for large language models." ICML 2023.
> [2] Kirchenbauer, John, et al. "On the reliability of watermarks for large language models." ICLR 2024.
> [3] https://github.com/jwkirchenbauer/lm-watermarking/blob/main/watermark_processor.py

---

> > ### Author Response · Authors · 2024-11-23
> > **A Gentle Reminder**
> >
> > We sincerely thank you for your valuable comments. Please let us know if our responses have adequately addressed your concerns. We deeply appreciate your feedback and view this as a valuable opportunity to improve our work. We would be very grateful if you could kindly share any feedback on our response.

---

> > > ### Author Response · Authors · 2024-12-01
> > >
> > > Thank you once again for your feedback! As the discussion period is nearing its end, we kindly ask if our response has adequately addressed your concerns. We sincerely appreciate your valuable suggestions and insights.

---

> > > > ### Comment · Reviewer_4ffB · 2024-12-02
> > > >
> > > > I thank the authors' detailed clarifications, which have addressed several of my concerns. However, I still have reservations about the manuscript's contribution, particularly regarding its significance and novelty. I, therefore, decided to keep my current score.
> > > >
> > > > **Significance:** The problem of watermarking for order-agnostic LLMs is not as challenging as the authors have claimed. In fact, there are simpler schemes that could be readily implemented.
> > > >
> > > > **Novelty:** The proposed technique can be viewed as a somewhat incremental extension of n-gram-based approaches to the order-agnostic setting.

---

> > > > > ### Author Response · Authors · 2024-12-02
> > > > > **Official Comment by Authors**
> > > > >
> > > > > We thank you for your feedback.
> > > > >
> > > > > > The problem of watermarking for order-agnostic LLMs is not as challenging as the authors have claimed. In fact, there are simpler schemes that could be readily implemented.
> > > > >
> > > > > We agree simpler schemes like Unigram can be readily implemented, however, we provide comprehensive experimental evidence that patternmark outperformed those simpler schemes. If there are other "simpler schemes" that we missed in our paper, we kindly ask the reviewer to point out the corresponding reference. Based on the current experimental evidence, we believe our results are significant.
> > > > >
> > > > > > The proposed technique can be viewed as a somewhat incremental extension of n-gram-based approaches to the order-agnostic setting.
> > > > >
> > > > > We respectfully disagree. First of all, **our method are not the extension of the n-gram-based approaches**. Actually, we never use the preceding n-gram as the watermark key, and thus our method is irrelevant with the n-gram-based apporaches.

---

### Official Review · Reviewer_cxkP · 2024-11-01

**Soundness:** 3
**Presentation:** 4
**Contribution:** 3
**Rating:** 6
**Confidence:** 3

**Summary:**

This paper proposes PATTERN-MARK, a watermarking method to label the output of order-agnostic language models (LMs). The authors developed a Markov-chain-based watermark generator to produce watermark key sequences, then assigned the keys one by one to the generated tokens to adjust their sampling probabilities. During detection, they first recover the key sequence from the suspected text and verify the watermark through hypothesis testing.

**Strengths:**

+） This paper is well-written and presents its ideas clearly.

+）This paper focuses on watermarking order-agnostic LMs, which, to the best of my knowledge, has not been considered in the existing literature.

+) This paper proposes an effective strategy to watermark order-agnostic LMs by embedding watermarks within the relationships between adjacent words.

**Weaknesses:**

-）I think the protein generation task is not suitable for experiments, as it may not be able to identify important unknown protein architectures.

**Questions:**

For the translation task, how can we identify whether a given text is generated by order-agnostic LMs or sequential LMs? This is critical for accurate detection.

---

> ### Author Response · Authors · 2024-11-13
> **Response to Reviewer cxkP**
>
> Thank you for acknowledging the novelty and effectiveness of our proposed methods. We greatly appreciate your valuable feedback. Below, we address your concerns:
>
> > W: I think the protein generation task is not suitable for experiments, as it may not be able to identify important unknown protein architectures.
>
> A: Our task is to detect whether a protein was generated by our watermarked model. All proteins generated by our model will be watermarked, though the true positive rate may not reach 100%.
>
> For proteins that are not generated by our watermarked model, for instance, some unknown proteins as you mentioned, we provide a provable theoretical false positive rate under the null hypothesis as described in Algorithm 3 and Algorithm 4. Thus we believe the protein generation task is well-suited for our experiment.
>
>
> > Q: For the translation task, how can we identify whether a given text is generated by order-agnostic LMs or sequential LMs? This is critical for accurate detection.
>
> A: We do not need to identify this. In current watermarking architectures, a service provider publishes their model and then detects whether a given output was generated by the specific released model. Therefore, our task is simply to verify whether the given text was generated by our watermarked, order-agnostic language model.
>
>
> Thank you again and if you have any further concerns, we are happy to discuss them with you.

---

> > ### Author Response · Authors · 2024-11-23
> > **A Gentle Reminder**
> >
> > We sincerely thank you for your valuable comments. Please let us know if our responses have adequately addressed your concerns. We deeply appreciate your feedback and view this as a valuable opportunity to improve our work. We would be very grateful if you could kindly share any feedback on our response.

---

> > > ### Author Response · Authors · 2024-12-01
> > >
> > > Thank you once again for your feedback! As the discussion period is nearing its end, we kindly ask if our response has adequately addressed your concerns. We sincerely appreciate your valuable suggestions and insights.

---

> > > > ### Comment · Reviewer_cxkP · 2024-12-03
> > > >
> > > > Thank you for your response, which addresses my concerns. I decide to maintain my score

---

### Official Review · Reviewer_gkea · 2024-11-02

**Soundness:** 2
**Presentation:** 2
**Contribution:** 2
**Rating:** 6
**Confidence:** 3

**Summary:**

This paper proposes a watermarking scheme for order-agnostic language models based on the proposed pattern-mark and hypothesis test. The method generates key sequences through a Markov-chain-based key generator, improves the probability of sampling from the sub-vocabulary corresponding to each key, and detects key sequences of specific patterns to calculate the false positive rate using hypothesis test. Compared with other watermarking methods, the method proposed in this paper shows superiority in protein generation and machine translation.

**Strengths:**

1.	This paper is well-organized and well-written.
2.	The discussion part of the paper provides a good explanation of the motivation for the method.

**Weaknesses:**

1.	The paper does not detail how the vocabulary set is divided. Splitting the vocabulary will inevitably affect the original probability distribution, resulting in a decrease in output quality. In addition, improper vocabulary segmentation may lead to grammatical errors in the generated sentences, such as incorrectly connecting the verb after the preposition. Is the part of speech considered when dividing the vocabulary?
2.	The probability outputs of language models often exhibit high probabilities for certain tokens while other tokens have much smaller probabilities, sometimes approaching zero. Although the factor used in Equation (1) aims to increase the probabilities of tokens in the sub-vocabulary, this amplification factor does not seem sufficient to bridge the gap between low-probability and high-probability tokens. In other words, if the vocabulary segmentation is not reasonable, it may not effectively enhance the sampling probability for specific sub-vocabularies. Particularly, when there are many sub-vocabularies, they may consist entirely of low-probability tokens. Was the original probability output considered when segmenting the vocabulary?
3.	The paper does not provide detailed explanations on how to obtain the set of target patterns. The target patterns should accurately reflect the characteristics of the specific key sequence.
4.	The comparative experiments in the experimental part of the paper are insufficient. This paper only compares the methods of two papers.

**Questions:**

NA

---

> ### Author Response · Authors · 2024-11-13
> **Response to Reviewer gkea (1/2)**
>
> Thank you for your time and valuable suggestions and we are grateful for your constructive feedback. Below, we provide a detailed response to address your concerns.
>
>
> > W1 a): The paper does not detail how the vocabulary set is divided. Splitting the vocabulary will inevitably affect the original probability distribution, resulting in a decrease in output quality.
>
> A: We randomly split the vocabulary, acknowledging that this approach inevitably affects the original probability distribution. However, this method is commonly used in many prevailing techniques [1,2,3] and is also the default setting in various real-world applications, such as the watermarking implementations in Hugging Face [4].
>
> While this probability distribution bias does result in a decrease in output quality, we emphasize that there is an inherent trade-off between watermarking strength and output quality. To achieve a stronger and more robust watermark, some sacrifice in output quality is unavoidable for order-agnostic models. As demonstrated in Figure 3, our method delivers the highest-quality outputs at the same watermarking strength compared to all baseline methods.
>
> > W1 b) In addition, improper vocabulary segmentation may lead to grammatical errors in the generated sentences, such as incorrectly connecting the verb after the preposition. Is the part of speech considered when dividing the vocabulary?
>
> The random segmentation does not lead to significant issues like grammatical errors, as we only linearly increase the probability of certain sub-vocabulary (multiplied by $e^{\delta}$), as described in Equation (1). Severe problems, such as grammatical errors, remain rare, as the probability change is minimal (e.g., from 1e-5 to 2e-5), and the output quality does not degrade significantly, which is also verified in previous works [1,5].
>
> > W2: The probability outputs of language models often exhibit high probabilities for certain tokens while other tokens have much smaller probabilities, sometimes approaching zero. Although the factor used in Equation (1) aims to increase the probabilities of tokens in the sub-vocabulary, this amplification factor does not seem sufficient to bridge the gap between low-probability and high-probability tokens. In other words, if the vocabulary segmentation is not reasonable, it may not effectively enhance the sampling probability for specific sub-vocabularies. Particularly, when there are many sub-vocabularies, they may consist entirely of low-probability tokens. Was the original probability output considered when segmenting the vocabulary?
>
> A:  We use random splitting and Equation (1), which are common techniques employed by many previous studies [1,2,3,5].
>
> When generating a specific token, there are instances where the sub-vocabulary may consist entirely of low-probability tokens, leading to the potential for an incorrect key to be recovered in Algorithm 2. However, we do not require all keys to be correctly recovered, as our detection algorithm is designed to be robust to such errors.
>
>
> Additionally, when generating multiple sentences, random splitting typically results in nearly equal probabilities for tokens to fall into different splits (around 0.5) across the vocabulary. To support this claim, we collect outputs from the non-watermarked model on the machine translation task and generate results using our split as well as five additional random splits. The results are shown below.
>
> |               | $P(x \in V_1)$ | $P(x \in V_2)$ |
> |---------------|:--------------:|:--------------:|
> | our split     |     48.13%     |     51.87%     |
> | random split1 |     48.58%     |     51.42%     |
> | random split2 |     47.51%     |     52.49%     |
> | random split3 |     49.34%     |     50.66%     |
> | random split4 |     49.95%     |     50.05%     |
> | random split5 |     49.59%     |     50.41%     |

---

> > ### Author Response · Authors · 2024-11-13
> > **Response to Reviewer gkea (2/2)**
> >
> > > W3: The paper does not provide detailed explanations on how to obtain the set of target patterns. The target patterns should accurately reflect the characteristics of the specific key sequence.
> >
> > A: The key patterns we used in our experiments are $T = $ {$k_1k_2k_1\ldots, k_2k_1k_2\ldots$}, where $k_1$ and $k_2$ appear alternately. It is clear that, given $a_{1,1}< a_{1,2}$, $ k_1$ and $k_2$ are more likely to occur alternatively.
> >
> > The length of the patterns, denoted as $m$ is a hyperparameter. We discuss the selection of $m$ in Section 4.5 (Lines 459-469), where we explain that a larger $m$ improves detection efficiency but also increases sensitivity to errors during key sequence recovery.
> >
> >
> > > W4: The comparative experiments in the experimental part of the paper are insufficient. This paper only compares the methods of two papers.
> >
> > A: We apologize for the limited number of baselines. However, as we are the first method specifically designed for order-agnostic language models, to the best of our knowledge, these are currently the only methods that can be effectively transferred and applied to such models. We would be happy to compare the performance of other watermarking schemes on order-agnostic language models if you could kindly suggest any for us.
> >
> >
> >
> > Thank you again and we would be more than happy to discuss any further questions or concerns you may have.
> >
> >
> > [1] Kirchenbauer, John, et al. "A watermark for large language models." ICML 2023. Outstanding Paper Award.
> > [2] Kirchenbauer, John, et al. "On the reliability of watermarks for large language models." ICLR 2024.
> > [3] Zhao, Xuandong, et al. "Provable robust watermarking for ai-generated text." ICLR 2024.
> > [4] https://huggingface.co/docs/transformers/en/generation_strategies#watermarking.
> > [5] Piet, Julien, et al. "Mark my words: Analyzing and evaluating language model watermarks." arXiv preprint arXiv:2312.00273 (2023).

---

> > > ### Comment · Reviewer_gkea · 2024-11-17
> > > **Response to Authors**
> > >
> > > Thanks for the detailed rebuttal. Most of my concerns have been addressed. I will raise my score.

---

> > > > ### Author Response · Authors · 2024-11-17
> > > > **Thank You!**
> > > >
> > > > We deeply thank you for taking the time to review our rebuttal and for acknowledging the clarifications we provided. We truly appreciate your thoughtful feedback throughout the review process and are glad we could address your concerns. Your valuable time and constructive comments are highly important to us!

---

### Official Review · Reviewer_h65u · 2024-11-08

**Soundness:** 3
**Presentation:** 2
**Contribution:** 1
**Rating:** 5
**Confidence:** 5

**Summary:**

The paper presents a watermarking method tailored for order-agnostic language models (LMs), which generate content in a non-sequential manner. The approach utilizes a Markov-chain-based key sequence to embed identifiable patterns within the generated content, enabling effective watermarking. Additionally, a statistical, pattern-based detection algorithm is employed for watermark verification. The authors also introduce a dynamic programming algorithm that optimizes the detection process by reducing its time complexity, enhancing the method's practical efficiency.

**Strengths:**

1. The paper effectively addresses the challenge of watermarking order-agnostic language models (LMs) by introducing a Markov-chain-based key sequence approach that overcomes the limitations inherent in traditional sequential watermarking methods.
2. The inclusion of a dynamic programming algorithm to optimize the detection process by significantly reduces the time complexity, thereby improving the practical feasibility of the proposed approach.
3. The proposed method enhanced detection accuracy and robustness, as well as the LMs output quality, in comparison to baseline methods.

**Weaknesses:**

1. The reliance on an alternating key sequence pattern introduces a potential vulnerability, as it may be more easily detected and disrupted by adversaries. Should the specific pattern structure (e.g., alternating keys) be identified, adversaries could develop targeted strategies to either erase or replicate the watermark. Incorporating more complex or adaptive key sequence strategies could enhance the method's robustness against such targeted disruptions.
2. The paper lacks a thorough discussion on why the proposed method, which utilizes alternating vocabulary splitting based on key sequences, outperforms global vocabulary splitting (e.g., the Unigram method) for watermarking order-agnostic LMs. Given that Unigram serves as a strong baseline, a detailed comparative analysis is needed to explain why the proposed approach achieves superior detection accuracy, robustness, and output quality despite both methods being context-independent.

**Questions:**

1. The authors should provide a detailed justification for why using an alternated vocabulary splitting strategy (the proposed method) offers advantages over a global vocabulary splitting approach (e.g., Unigram) in terms of output quality and watermark robustness in order-agnostic LMs, given that both methods are context-independent?
2. In Table 4, what does the term "attack strength ε" represent in the context of the ChatGPT paraphrasing attack? Additionally, how is this attack strength controlled or quantified during the experiments?
3. Could the authors clarify the mathematical meaning of \( e^\delta \) in Equation (1)?
4. The method is described as Markov-chain-based due to its key sequence generation process; however, the paper’s use of an alternating key sequence (e.g., a fixed pattern like \( k_1, k_2, k_1, k_2 \)) does not appear to leverage the stochastic properties of Markov chains. This seems potentially misleading, as the approach is more akin to an alternated 0/1 key sequence than a Markov-chain-based generation.
5. The proposed method employs fixed vocabulary splitting, which resembles the Unigram approach and may be easier to detect. What justification do the authors provide for the detectability and resilience of the proposed watermark against adversarial attempts to identify or remove it?

---

> ### Author Response · Authors · 2024-11-13
> **Response to Reviewer h65u (1/2)**
>
> Thank you for your valuable feedback and insightful comments. We greatly appreciate the time and effort that went into reviewing our manuscript.  Below, we address each of your comments in detail.
>
> > W1: The reliance on an alternating key sequence pattern introduces a potential vulnerability, as it may be more easily detected and disrupted by adversaries. Should the specific pattern structure (e.g., alternating keys) be identified, adversaries could develop targeted strategies to either erase or replicate the watermark. Incorporating more complex or adaptive key sequence strategies could enhance the method's robustness against such targeted disruptions.
>
> A: Our primary goal is to design a watermarking scheme that is both strong and minimizes distribution bias, while defense against specific attacks is not our main focus. Identifying effective watermarking schemes remains challenging in this field, and we did not find a straightforward way to attack our method to reveal any potential vulnerabilities.
>
> More importantly, we argue that our approach is more robust against such attacks compared to baseline methods. The baseline Unigram watermarking relies on a fixed green list, leading to a clear bias toward green list tokens and creating vulnerabilities. The Soft watermarking method directly uses a previous token to seed the random number generator, which makes the pattern $p(x_1|x_2)$ potentially inferable through a large number of queries. Instead, our method only introduces a dependency between underlying keys using a Markov chain, adding complexity to the task of reconstructing the indirect correlation between neighboring tokens. Furthermore, we can increase the values of $a_{1,1}$ and $a_{2,2}$ to introduce more stochasticity into the Markov chain, strengthening our defense capabilities. We can also design more complex Markov chains to further reduce potential vulnerabilities.
>
>
> > W2: The paper lacks a thorough discussion on why the proposed method, which utilizes alternating vocabulary splitting based on key sequences, outperforms global vocabulary splitting (e.g., the Unigram method) for watermarking order-agnostic LMs. Given that Unigram serves as a strong baseline, a detailed comparative analysis is needed to explain why the proposed approach achieves superior detection accuracy, robustness, and output quality despite both methods being context-independent.
>
> A:  Unigram uses a fixed red-green list to split the vocabulary and only increase the probability of green tokens, which means the output distribution is consistently biased toward the green tokens, i.e.,$ P(x_i \in green\ tokens) > P(x_i \in red\ token)$, which ultimately reduces output quality.
>
> In contrast, our Markov-based method sets $Q_1=Q_2=0.5$, $a_{1,1}=a_{2,2}$, $a_{1,2}=a_{2,1}$, ensuring that $P(x_i \in V_1)=P(x_i \in V_2), \forall i$, indicating a more balanced token distribution. Additionally, our approach uses alternating keys, which contribute to a more diverse output (as shown in Table 7) and, consequently, higher output quality.
>
> In terms of detection accuracy, Unigram can be viewed as a special case of our proposed Pattern-mark, where $Q_1=1$, $a_{1,1}=1$, and the target pattern is $\{k_1\}$, with the length $m$ is only 1. As shown in Figure 4, setting the pattern length to only 1 results in the loss of important information, leading to reduced detection accuracy.
>
> For robustness against paraphrase attacks, as our method has a stronger detection ability, it can still maintain relatively high robustness accuracy compared to the Unigram baseline.
>
> > Q1: The authors should provide a detailed justification for why using an alternated vocabulary splitting strategy (the proposed method) offers advantages over a global vocabulary splitting approach (e.g., Unigram) in terms of output quality and watermark robustness in order-agnostic LMs, given that both methods are context-independent?
>
> A:  Please refer to the response for W2, as the concerns are the same.
>
> > Q2: In Table 4, what does the term "attack strength $\epsilon$" represent in the context of the ChatGPT paraphrasing attack? Additionally, how is this attack strength controlled or quantified during the experiments?
>
> A: We are deeply sorry for the confusion. In our experiments, given the length of an output is $l$, we randomly select a subsequence whose length is $\epsilon l$ and use ChatGPT to paraphrase it.

---

> ### Author Response · Authors · 2024-11-13
> **Response to Reviewer h65u (2/2)**
>
> > Q3: Could the authors clarify the mathematical meaning of ( e^\delta ) in Equation (1)?
>
> A:  We follow the probability promotion strategy in Kirchenbauer et al. (2023) [1], which splits the token list into a red and a green list, then increases the logits of the green tokens by $\delta$. In order to transfer the token logits to the probability vector, the softmax function $\frac{e^{logits_k}}{\sum e^{logits_i}}$ is applied to the LM logits. As we increase the logits of green tokens by $\delta$, the green token probabilities will be $\frac{e^{logits_k+\delta}}{\sum_{red} e^{logits_i}+\sum_{green} e^{logits_i+\delta}}=\frac{e^{\delta}e^{logits_k}}{\sum_{red} e^{logits_i}+\sum_{green} e^{\delta}e^{logits_i}}$. That’s the mathematical meaning of $e^{\delta}$ in Equation (1).
>
>
> > Q4: The method is described as Markov-chain-based due to its key sequence generation process; however, the paper’s use of an alternating key sequence (e.g., a fixed pattern like ( k_1, k_2, k_1, k_2 )) does not appear to leverage the stochastic properties of Markov chains. This seems potentially misleading, as the approach is more akin to an alternated 0/1 key sequence than a Markov-chain-based generation.
>
> A: Our framework employs a Markov chain, with the transition matrix serving as a hyperparameter. Adjusting the transition matrix allows us to balance output diversity, robustness against potential attacks, and watermark strength. When using a smaller $a_{1,1}$, watermark strength is enhanced, as it differs more from the null hypothesis and is more likely to form the desired key pattern for detection. However, this may also reduce output diversity and robustness against potential identification or removal attacks.
>
> In our experiments, we show that using $a_{1,1}=0$, i.e., alternating the key sequence does not reduce the output quality too much (shown in Figure 5), and defense against identification or removal attacks is not our main concern, thus we choose $a_{1,1}=0$.
>
> In tasks where diversity is a priority or defense capability is essential, a non-zero $a_{1,1}$ can be selected.
>
>
> > Q5: The proposed method employs fixed vocabulary splitting, which resembles the Unigram approach and may be easier to detect. What justification do the authors provide for the detectability and resilience of the proposed watermark against adversarial attempts to identify or remove it?
>
> A: First of all, defense against watermark identifying and removing attacks remains a relatively unexplored area and is not the primary focus of this paper. Moreover, we do not observe any obvious vulnerabilities or straightforward methods to identify or remove our proposed watermark.
>
> Additionally, we argue that our approach is more robust against such attacks compared to baseline methods. The baseline Unigram watermarking relies on a fixed green list, leading to a clear bias toward green list tokens and creating vulnerabilities. The Soft watermarking method directly uses a previous token to seed the random number generator, which makes the pattern $p(x_1|x_2)$ potentially inferable through a large number of queries.
>
> Instead, our method only introduces a dependency between underlying keys using a Markov chain, adding complexity to the task of reconstructing the indirect correlation between neighboring tokens. Furthermore, we can increase the values of $a_{1,1}$ and $a_{2,2}$ to introduce more stochasticity into the Markov chain, strengthening our defense capabilities. Additionally, we can design more complex Markov chains to further reduce potential vulnerabilities.
>
>
>
> Thank you again for your thoughtful and constructive feedback. Please let us know if anything needs further clarification and we are more than happy to make additional improvements.
>
> [1] Kirchenbauer, John, et al. "A watermark for large language models." ICML 2023.

---

> > ### Author Response · Authors · 2024-11-23
> > **A Gentle Reminder**
> >
> > We sincerely thank you for your valuable comments. Please let us know if our responses have adequately addressed your concerns. We deeply appreciate your feedback and view this as a valuable opportunity to improve our work. We would be very grateful if you could kindly share any feedback on our response.

---

> ### Comment · Reviewer_h65u · 2024-11-26
>
> Thank you for your response. I still have some questions regarding the proposed method:
>
> 1. **Robustness:** Since watermark detection relies on the relative permutation of tokens to retrieve the watermark signal, could the method be particularly sensitive to modifications involving changes in token permutations?
>
> 2. **Security:** The proposed watermarking method raises security concerns due to the use of fixed vocabulary splitting. Specifically, an attacker could potentially recover the vocabulary splitting by analyzing token frequencies for odd and even indexed tokens when the watermark is guided with an alternative key sequence.
>
> 3. **Transition Matrix:** Figure 5 does not show a clear trend in generation diversity or BLEU scores when adjusting the transition matrix. Additionally, there are no further results on how the transition matrix impacts robustness, leaving the role of the transition matrix unclear.

---

> ### Author Response · Authors · 2024-11-26
> **Response to Follow-Up Questions**
>
> Thank you for your valuable feedback. We sincerely appreciate the time and effort you dedicated to reviewing our manuscript and rebuttal. Below, we address your remaining concerns:
>
>
> > Follow-up Q1: **Robustness**: Since watermark detection relies on the relative permutation of tokens to retrieve the watermark signal, could the method be particularly sensitive to modifications involving changes in token permutations?
>
> No, our method is not particularly sensitive to modifications involving changes in token permutations. Although our watermark detection requires the previous token information to retrieve the signal. However, this is also true for the baseline Soft Watermark, which uses the preceding token to reconstruct the watermark key. More importantly, leveraging preceding token information is a widely adopted practice in various prior works [1,2,3,4,5].
>
> Moreover, altering the relative order of tokens can significantly degrade content quality. For instance, given the sentence "Matt sat on the bench," rearranging the tokens arbitrarily—such as "on Matt sat the bench"—renders the sentence incoherent and difficult to understand.
>
> Regarding robustness against other modifications, we kindly point out that our experiments in Table 3 and Table 4 demonstrate that our proposed method exhibits stronger robustness compared to the baseline methods, with only a small performance drop under such modifications.
>
>
> > Follow-up Q2: **Security**: The proposed watermarking method raises security concerns due to the use of fixed vocabulary splitting. Specifically, an attacker could potentially recover the vocabulary splitting by analyzing token frequencies for odd and even indexed tokens when the watermark is guided with an alternative key sequence.
>
> Thank you for raising this concern. However, the attack method you proposed will not work. As mentioned in Line 370, our initial distribution is $Q=[0.5,0.5]$, ensuring that the key sequence for different outputs will vary. For example, for a sequence with three tokens, the corresponding key sequence could be $k_1k_2k_1$ or $k_2k_1k_2$, each with a 50% probability.
>
> As a result, a token $x_i$ at the same position $i$ will have $P(x_i \in V_1)=P(x_i \in V_2)=0.5$, ensuring that the token frequency at even and odd indices remains identical across multiple sequences. This makes it impossible to attack our method using token frequency analysis.
>
> Moreover, to further enhance defense against the proposed attack, we can set $a_{11}$ to a small non-zero value, such as $a_{11}=0.1$. This adjustment generates key sequences like $k_1 k_2 k_2 k_1 k_2$, where keys do not alternate strictly, further thwarting frequency analysis-based attacks
>
>
>
> > Follow-up Q3: **Transition Matrix**: Figure 5 does not show a clear trend in generation diversity or BLEU scores when adjusting the transition matrix.
>
>
> First, we would like to clarify the role of the transition matrix:
>
> Our detection algorithm relies on the occurrence of key patterns, and we select alternating keys as the target pattern, such as {$k_1k_2k_1k_2,k_2k_1k_2k_1$}. As discussed in Lines [260–267], a smaller $a_{11}$ (i.e., a large $a_{12}$ since $a_{11}+a_{12}=1$) encourages the generation of alternating keys. Therefore, reducing $a_{11}$ improves detection performance.
>
> Furthermore, as discussed in Lines [470–476], our results in Figure 5 show that adjusting the transition matrix has no significant effect on BLEU scores or pLDDT, which you also acknowledged. Hence, we can simply set $a_{11}=0$ and $a_{12}=1$ to achieve optimal detection performance.
>
>
> > Follow-up Q4: Additionally, there are no further results on how the transition matrix impacts robustness, leaving the role of the transition matrix unclear.
>
> In terms of robustness, increasing $a_{11}$ reduces the match between the generated key sequences and the target pattern (i.e., alternating keys), leading to a drop in robustness. Since Tables 4 and 5 already report the robustness results for the strongest setting of our method, we believe reporting robustness under different $a_{11}$ values would not provide additional meaningful insights.
>
>
> Thank you again for your feedback and we are more than happy to provide further clarification if you have any additional concerns.
>
> [1] Kirchenbauer, John, et al. "A watermark for large language models." ICML 2023.
> [2] Kirchenbauer, John, et al. "On the reliability of watermarks for large language models." ICLR 2024.
> [3] Hu, Zhengmian, et al. "Unbiased watermark for large language models." ICLR 2024.
> [4] Wu, Yihan, et al. "Dipmark: A stealthy, efficient and resilient watermark for large language models." ICML 2024.
> [5] Huo, Mingjia, et al. "Token-Specific Watermarking with Enhanced Detectability and Semantic Coherence for Large Language Models." ICML 2024.

---

> > ### Author Response · Authors · 2024-12-01
> >
> > Thank you once again for your feedback! As the discussion period is nearing its end, we kindly ask if our response has adequately addressed your concerns. We sincerely appreciate your valuable suggestions and insights.

---

### Meta-Review · Area_Chair_33aD · 2024-12-16

**Metareview:**

This paper proposes a watermark for order-agnostic LLMs. This was a borderline submission, with two reviewers suggesting "slightly above accept threshold" and two "slightly below". During the AC-reviewer discussion, we discussed the merits of the paper. The AC also read the paper and found that the issues raised by the reviewers are generally addressed in the work, including robustness to perturbations, improved performance compared to simple unigram methods, and improved security compared to baselines (random splits of the vocab appear to be more secure than fixed ones). Based on this, I can recommend acceptance.

**Additional Comments On Reviewer Discussion:**

There was a long discussion, already summarized above.

---

### Decision · Program_Chairs · 2025-01-22

Accept (Poster)